# Advances in Catalysts for Water–Gas Shift Reaction Using Waste-Derived Synthesis Gas

Ru-Ri Lee [1,†], I-Jeong Jeon [1,†], Won-Jun Jang [2,*], Hyun-Seog Roh [3,*] and Jae-Oh Shim [1,4,*]

1. Department of Chemical Engineering, Wonkwang University, 460 Iksan-daero, Iksan-si 54538, Republic of Korea
2. Department of Environmental and Energy Engineering, Kyungnam University, 7 Kyungnamdaehak-ro, Changwon-si 51767, Republic of Korea
3. Department of Environmental and Energy Engineering, Yonsei University, 1 Yonseidae-gil, Wonju-si 26493, Republic of Korea
4. Nanoscale Environmental Sciences and Technology Institute, Wonkwang University, 460 Iksan-daero, Iksan-si 54538, Republic of Korea
* Correspondence: wjjang@kyungnam.ac.kr (W.-J.J.); hsroh@yonsei.ac.kr (H.-S.R.); joshim85@wku.ac.kr (J.-O.S.)
† These authors contributed equally to this work.

**Abstract:** Hydrogen is mainly produced by steam reforming of fossil fuels. Thus, research has been continuously conducted to produce hydrogen by replacing fossil fuels. Among various alternative resources, waste is attracting attention as it can produce hydrogen while reducing the amount of landfill and incineration. In order to produce hydrogen from waste, the water–gas shift reaction is one of the essential processes. However, syngas obtained by gasifying waste has a higher CO concentration than syngas produced by steam reforming of fossil fuels, and therefore, it is essential to develop a suitable catalyst. Research on developing a catalyst for producing hydrogen from waste has been conducted for the past decade. This study introduces various catalysts developed and provides basic knowledge necessary for the rational design of catalysts for producing hydrogen from waste-derived syngas.

**Keywords:** hydrogen; water–gas shift reaction; catalyst; waste-derived syngas

## 1. Introduction

The amount of waste generated globally is gradually increasing due to rising population, urbanization, and industrialization [1–9]. In 2020, eight billion people generated 2.5 billion tons of wastes, and this amount is predicted to rise to 5.9 billion tons by 2050 [8,10]. As a result, there is a growing interest in environmental sustainability research. Wastes are processed using incineration, landfill, and other techniques. When wastes are processed in this manner, there is an issue of insufficient landfill sites and air pollution caused by the gas generated during incineration. Thus, substantial waste treatment research is ongoing. One example is the process of transforming waste into hydrogen [4,11]. Through the process of gasification, purification, water–gas shift (WGS), and separation, the waste-to-energy conversion (WtE) employing waste can create high-purity hydrogen [12]. Among these processes, the WGS reaction is a key step for producing high-purity hydrogen from waste [4,11].

Considering the thermodynamic characteristics of the WGS reaction, this reaction is operated in two stages: the high-temperature shift (HTS) reaction carried out at a temperature range of 350–500 °C and the low-temperature shift (LTS) reaction carried out at a temperature range of 200–250 °C [12–19]. The Fe-Cr catalyst is used as a commercial catalyst in the HTS reaction, while the Cu-Zn-Al catalyst is used as a commercial catalyst in the LTS reaction [11,12,20–26]. However, there is a downside in the case of the Fe-Cr catalyst utilized in the HTS process in that it produces hexavalent chromium, which is a

carcinogen [27–30]. To address these issues, research into using a catalyst other than the Fe-Cr catalyst is under way. Various studies are being undertaken to find catalysts for the WGS reaction in the WtE process.

Active metals have a significant impact on the catalysts utilized in the WGS process. There are two categories of materials utilized as active metals: noble and non-noble metals. According to current research, non-noble metal catalysts such as Fe-based, Cu-based, Ni-based, and Co-based catalysts, as well as noble metal catalysts such as Pt-based catalysts, are utilized in the WtE process. In the HTS reaction, the Fe-based catalyst reduces $Fe_2O_3$ to the active species $Fe_3O_4$ [31–33]. Cr is a substance that prevents sintering of $Fe_3O_4$, enhances stability in the Fe-Cr catalyst, and modifies catalytic behavior of $Fe_3O_4$ in the Fe-Cr catalyst, which is a commercial catalyst [34–39]. However, as previously stated, $Cr^{6+}$ materials have the potential to pollute the environment [40–43]. Al, Cu, and Ni metals have recently received a lot of interest owing to their strong activity in the HTS reaction. The physicochemical properties and synergistic effects of Fe-based catalysts employing metal oxides as supports have been explored, and numerous investigations are being undertaken on techniques that do not utilize Cr [11–13,28,44]. Commercially, Cu-based catalysts are used in the LTS process, and Cu catalysts with oxide supports have shown significant activity in the WGS reaction [27]. The Cu catalyst, on the other hand, has the problem of quickly deactivating at high temperatures, making it unsuitable for the HTS reaction. Consequently, many researchers have investigated thermally stable supports, such as $CeO_2$ and $Al_2O_3$, that may increase the performance of the catalyst in order to employ the Cu catalyst in the HTS process [27,45–48]. Oxide-supported Cu-based catalysts have received significant attention over the last few decades due to their high activity in WGS reactions [27]. In the HTS reaction, the Co-based catalyst interacts with $Co^0$ as an active species [1,6,49,50]. Lee et al. researched the $Co/CeO_2$ catalyst in anticipation of $Co_3O_4$ activity in the HTS process, based on the fact that $Co_3O_4$ was chosen as an effective catalyst for oxidizing CO to $CO_2$ [1,51]. There has been a lot of research utilizing Co as an active metal and $CeO_2$ as a support, as well as work on adding metal components to improve catalysts and boost catalytic activity [1,6,52]. Pt-based catalysts have been investigated with an emphasis on the sulfur resistance necessary in the industrial WtE process [53]. In the WGS process, sulfur acts as a poison to the catalyst, becoming highly adsorbed on the active site of the catalyst, thus inactivating it [54–59]. Therefore, it is crucial to develop a catalyst capable of removing sulfur, and the sulfur resistance and regeneration of catalyst impact catalyst activity. Promoters also play an important role in the WGS reaction. Barium enhances the strong metal–support interaction between active Cu metal and Ce-Al support, thereby improving the catalytic performance for the HTS reaction. Barium also enhances the sintering resistance of metallic cobalt and improves the reducibility of the Co-based catalyst. Calcium increases the formation of oxygen vacancies for the Ni-$CeO_2$ catalyst, which are important for the WGS reaction.

This review article focuses on catalysts for hydrogen production from waste-derived synthesis gas. Among WGS reaction catalysts for hydrogen production from combustible municipal solid waste, Fe-, Cu-, Ni-, Co-, and Pt-based catalysts, which are known to have outstanding performance and features, will be examined. Furthermore, the simple reducibility of the Fe-based catalyst, which considerably influences the performance in the WGS process, the degree of dispersion and redox capacity of Cu, the oxygen storage capacity of Ni and Co catalysts supported on $CeO_2$, and a noble metal-based catalyst are all described in detail. The objective of this study is to obtain a better knowledge of the critical factors that must be considered while designing catalysts in changing conditions.

## 2. Overview

### 2.1. Brief History of Water–Gas Shift Reaction and Catalysts

"Water gas" is a mixture of hydrogen and carbon monoxide which is typically produced by the reaction of hydrocarbons such as natural gas, coal, waste, and biomass with oxidizing agents such as steam, oxygen, or carbon dioxide [60]. The water–gas shift (WGS)

reaction was used to produce hydrogen for the synthesis of $NH_3$, which is well-known for the Haber process [61–63]. During the recent several decades, the WGS reaction has received great attention again in parallel with the utilization of hydrogen as a clean and sustainable energy carrier [64,65]. Currently, the steam reforming of natural gas (SRNG) is one of the most economical ways to produce hydrogen [66–68]. The hydrogen production via SRNG comprises desulfurization, reforming, CO conversion (WGS), and CO elimination. Herein, the CO conversion step means the WGS reaction, which is a process to convert CO with $H_2O$ into $CO_2$ and $H_2$ ($CO + H_2O = H_2 + CO_2$, $\Delta H = -41.2$ kJ/mol). The industrial WGS process consists of two stages at two temperature ranges considering the thermodynamic and kinetic aspects: high-temperature shift (HTS, 350~500 °C) and low-temperature shift (LTS, 200~250 °C) [12–19]. $Fe_2O_3$-$Cr_2O_3$ and $CuO$-$ZnO$-$Al_2O_3$ catalysts have been commercially used for the sequential process of HTS and LTS, respectively.

Conventional commercial HTS catalysts are composed of ca. 80–90% of $Fe_2O_3$, 8–10% of $Cr_2O_3$, and the balance being promoter and stabilizer such as $CuO$, $Al_2O_3$, alkali, $MgO$, $ZnO$, etc. [69,70]. $Fe_3O_4$(magnetite), which is formed by the partial reduction of $\alpha$-$Fe_2O_3$(hematite), is an active phase for the WGS reaction in the high-temperature range [33,71]. $Cr_2O_3$ mainly functions as a textual promoter to prevent the sintering of active $Fe_3O_4$ phase and loss of surface area during start-up and operation [33,71]. $Fe_2O_3$-$Cr_2O_3$ catalysts have been commercially used for more than 70 years despite low performance at a low temperature because they showed the excellent catalytic performance at a high temperature [72].

Commercial LTS catalysts are a mixture of ca. 60–70% of $CuO$, 20–30% of $ZnO$, 10% of $Al_2O_3$, and the balance being promoter and stabilizer such as $Cr_2O_3$, Cs, or $MnO$ [73,74]. In general, Cu metal crystallites are known to be an active species for the WGS reaction in the low-temperature range [74,75]. $ZnO$ and $Cr_2O_3$ provide the structural support for the catalyst and $Al_2O_3$, which is inactive for the WGS reaction, enhances the Cu dispersion, and minimizes the collapse of pellet [76]. Cs helps to improve the selectivity to the WGS reaction. The commercial LTS catalysts were prone to Cu sintering and this resulted in the subsequent loss of Cu surface area [77,78]. Moreover, the catalysts were sensitive to temperature and pyrophoric in air [79,80]. In spite of these problems, the catalytic performance of the commercial LTS catalysts in a low temperature is excellent and comparable to that of noble-metal-based catalysts [77,81]. Thus, $Cu$-$ZnO$-$Al_2O_3$ catalysts are commercially still in use.

However, both commercial HTS and LTS catalysts were designed for the conversion of synthesis gas that was produced by the steam reforming of natural gas. Recently, the optimization of WGS catalysts is required in parallel with the different compositions of synthesis gas produced from various resources such as waste, biomass, and coal [82–84]. Many researchers have focused on the production of hydrogen using waste due to the lack of hydrocarbon resources [85–87]. Thus, the WGS catalysts have been widely studied to produce hydrogen from waste-derived synthesis gas. The following section will discuss in detail various catalysts of Fe-, Cu-, Ni-, Co-, and Pt-based catalysts and their catalytic properties.

## 2.2. The Composition of Waste-Derived Synthesis Gas

Table 1 summarizes the composition of the waste-derived syngas utilized in the HTS process. Most catalysts had gas concentrations of 38.2 vol% CO, 21.5 vol% $CO_2$, 2.3 vol% $CH_4$, 29.2 vol% $H_2$, and 8.8 vol% $N_2$. The formula for $H_2O$ was $H_2O/(CH_4 + CO + CO_2)$ = 2.0 [1,11,12,28]. Furthermore, the synthesis gas composition of the Fe-, Cu-, Co-, and Pt-based catalysts, including $H_2O$, was determined to be 55.20 vol% $H_2O$, 17.02 vol% CO, 9.55 vol% $CO_2$, 1.03 vol% $CH_4$, 13.14 vol% $H_2$, and 4.06 vol% $N_2$. Furthermore, omitting $H_2O$, the gas composition was determined to be 37.99 vol% CO, 21.32 vol% $CO_2$, 2.30 vol% $CH_4$, 29.33 vol% $H_2$, and 9.06 vol% $N_2$ [6,8,13,48,53]. The Cu-based catalyst's gas concentration was 38.0 vol% CO, 21.3 vol% $CO_2$, 2.3 vol% $CH_4$, 29.3 vol% $H_2$, and 9.1 vol% $N_2$ [26,27]. The Co-based and Pt-based catalysts' syngas compositions were 39.7 vol% CO,

21.5 vol% $CO_2$, 2.35 vol% $CH_4$, 27.05 vol% $H_2$, and 9.40 vol% $N_2$ [57,88]. The Co-based and Ni-based catalysts had syngas compositions of 37.87 vol% CO, 21.47 vol% $CO_2$, 2.30 vol% $CH_4$, 29.31 vol% $H_2$, and 9.05 vol% $N_2$ [87,89].

**Table 1.** Composition of gaseous mixture used in HT-WGS reaction.

| Catalyst | CO (vol%) | CO$_2$ (vol%) | CH$_4$ (vol%) | H$_2$ (vol%) | N$_2$ (vol%) | H$_2$O (vol%) | Ref. |
|---|---|---|---|---|---|---|---|
| Fe-based | 38.2 | 21.5 | 2.3 | 29.2 | 8.8 | - | [11,12,28] |
| | 17.02 | 9.55 | 1.03 | 13.14 | 4.06 | 55.20 | [13] |
| | 37.99 | 21.32 | 2.30 | 29.33 | 9.06 | - | [13] |
| Cu-based | 38.0 | 21.3 | 2.3 | 29.3 | 9.1 | - | [26,27] |
| | 37.99 | 21.28 | 2.31 | 29.34 | 9.08 | - | [48] |
| Co-based | 38.2 | 21.5 | 2.3 | 29.2 | 8.8 | - | [1] |
| | 17.02 | 9.55 | 1.03 | 13.14 | 4.06 | 55.20 | [6,8,53] |
| | 37.99 | 21.32 | 2.30 | 29.33 | 9.06 | - | [6,8,53] |
| | 39.70 | 21.50 | 2.35 | 27.05 | 9.40 | - | [88] |
| | 37.87 | 21.47 | 2.30 | 29.31 | 9.05 | - | [89] |
| Pt-based | 37.99 | 21.28 | 2.31 | 29.34 | 9.08 | - | [59] |
| | 39.70 | 21.50 | 2.35 | 27.05 | 9.40 | - | [57] |
| Ni-based | 37.87 | 21.47 | 2.30 | 29.31 | 9.05 | - | [87] |

## 3. Overview of Catalyst Results

### 3.1. Fe-Based Catalyst (Easy Reducibility of Fe$_2$O$_3$)

Fe-based catalysts are typically used for the conversion of synthesis gas derived from waste at a high temperature. Since the carcinogenic nature of hexavalent chromium ($Cr^{6+}$) was reported, the development of a Cr-free catalyst is a matter of great significance to replace a commercial Fe-Cr catalyst. Attempts at adding the substitute materials or using various preparation methods have been reported to obtain a small size of particle and to improve the reducibility of Fe, which is one possible way to enhance the catalytic activity resulting from the rapid redox cycle between $Fe^{2+}$ and $Fe^{3+}$. In this section, four papers related to Fe-based catalysts are summarized and distinguished by naming [FAC], [FACP], [FACS], and [CFMA] in front of descriptions for each characteristic analysis.

**[FAC]** Among these, studies have demonstrated that a modest amount of metal oxide doped into the Fe/Al catalyst increases catalytic activity. Fe/Al catalysts, Fe/Al/Cu catalysts doped with metal oxides (Cu, Ni), and Fe/Al/Ni catalysts were thus compared. The Fe/Al/Cu catalyst achieved the 80.9% (450 °C) of CO conversion and 100% $CO_2$ selectivity, which is a higher value than that of the Fe/Al and Fe/Al/Ni catalysts. The high catalytic performance of the Fe/Al/Cu catalyst is mainly attributed to the synergistic effect between Fe and Cu, which results from enhanced reducibility [12]. **[FACP]** After establishing that doping, the Fe/Al catalyst with Cu, a metal oxide, improved catalytic activity; the effect of the catalyst manufacturing process on catalytic activity was examined. Co-precipitation (CP), sol-gel (SG), and impregnation (IM) were used to synthesize Fe/Al/Cu catalysts. It was found that the CP-method-prepared catalyst had the maximum activity. The high activity of catalysts was found to be attributable to its large BET surface area, small crystal size of $Fe_3O_4$, easy reducibility, and production of reduced Cu species [11]. **[FACS]** The production amount per batch of the Fe/Al/Cu catalyst prepared by the co-precipitation method was increased. Based on the FAC-PC-1 catalyst, which produced 2 g of catalyst per batch, the catalyst with a three-fold increase in production (6 g) was named FAC-PC-3, the catalyst with a five-fold increase (10 g) was named FAC-PC-5, and the catalyst with a ten-fold increase (20 g) was named FAC-PC-10. As a consequence, when three-fold scaled up, it demonstrated good activity comparable to the current catalytic process. Thus, based on the FAC-PC-3 catalyst, the FAC-PC-3-240 catalyst was developed, which increased the production amount by 40 times. Only the $Fe_3O_4$ crystallite size rose somewhat as compared

to the FAC-PC-3 catalyst, while the rest of the characteristics remained consistent [28].
**[FAC]** Through the characterization results of the used catalysts summarized in Table 2, the cause of the high activity of the Fe/Al/Cu catalyst was elucidated. The catalyst exhibited higher activity when the BET surface area of the used catalyst was larger and the crystal size of $Fe_3O_4$ was smaller. In comparison to the Fe/Al and Fe/Al/Ni catalysts, the Fe/Al/Cu catalyst had the largest surface area and the lowest $Fe_3O_4$ crystallite size [12]. **[FACP]** The FAC catalyst exhibited the highest BET surface area and the smallest $Fe_3O_4$ crystal size when prepared using the co-precipitation method [11]. **[FACS]** The FAC-PC-3 catalyst, which increased the production amount by three times compared to the FAC-PC-1 catalyst, exhibited almost the same BET surface area and $Fe_3O_4$ crystal size as the FAC-PC-1 catalyst. Furthermore, when compared to other catalysts, the FAC-PC-3 catalyst had a higher Cu dispersion of 5.7%. This is comparable to the previous FAC-PC-1 catalyst's Cu dispersion of 5.9%, and the FAC-PC-240 catalyst's Cu dispersion of 5.6% [28].

**Table 2.** Characteristics of Fe-based catalysts.

| Catalyst | BET Surface Area (m²/g) [a] | | Crystallite Size (nm) [b] | | Ref. |
|---|---|---|---|---|---|
| | Fresh | Used | Fresh ($Fe_2O_3$) | Used ($Fe_3O_4$) | |
| Fe/Al | 56.6 | 12.5 | 18.4 | 24.4 | [12] |
| Fe/Al/Cu | 73.0 | 20.3 | 17.3 | 17.9 | [12] |
| Fe/Al/Ni | 81.4 | 11.2 | 14.9 | 20.6 | [12] |
| FAC-CP | 165.1 | 32.6 | N.A. [c] | 18.1 | [11] |
| FAC-SG | 104.9 | 20.1 | - | 23.0 | [11] |
| FAC-IM | 73.0 | 15.5 | 17.3 | 27.4 | [11] |
| FAC-PC-1 (Cu dispersion: 5.9%) | 168 | - | - | 13.4 | [28] |
| FAC-PC-3 (Cu dispersion: 5.7%) | 165 | - | - | 13.7 | [28] |
| FAC-PC-5 (Cu dispersion: 4.7%) | 132 | - | - | 16.6 | [28] |
| FAC-PC-10 (Cu dispersion: 2.8%) | 60 | - | - | 22.3 | [28] |
| FAC-PC-3-240 (Cu dispersion: 5.6%) | 166 | - | - | 15.9 | [28] |

[a] Estimated from the $N_2$ adsorption isotherm at $-196\ ^\circ C$. [b] Calculated from (2 0 0) peak of metallic Cu using the Scherrer equation. [c] Not available due to very broad and weak XRD peaks.

**[FACS]** The FAC-PC-3 catalyst, which had been grown up three-fold, exhibited unique particles identical to the previous FAC-PC-1 catalyst. SAED pattern analysis verified this, as shown in Figure 1. Unlike other catalysts, which exhibited $\alpha$-$Fe_2O_3$ (JCPDS #33-0664) at (110), (113), (202), and (116) in SAED pattern analysis, FAC-PC-3 showed $\gamma$-$Fe_2O_3$ (JCPDS #39-1346) at (311) and (440) [28]. In the WGS reaction, the production of $\gamma$-$Fe_2O_3$ is important to form active $Fe_3O_4$ species. $Fe_3O_4$ is widely recognized as the active phase of the target reaction in the case of Fe-based catalysts, and it has an inverted spinel structure represented by $[Fe^{3+}]^{tetra}[Fe^{3+}Fe^{2+}]^{octa}O_4$. In contrast to completely oxidized $\alpha$-$Fe_2O_3$, which contains only octahedral $FeO_6$, $\gamma$-$Fe_2O_3$ can be quickly converted into $Fe_3O_4$. This is because $\gamma$-$Fe_2O_3$ and $Fe_3O_4$ have structural similarities with a twist of tetrahedral $FeO_4$ or octahedral $FeO_6$ [28]. $H_2$-TPR analysis was performed to assess the catalyst's reducibility, as shown in Table 3. **[FAC]** All catalysts in the Fe/Al/Cu family displayed three reduction temperature ranges as a consequence of TPR (Table 3). At 178, 325, and $660\ ^\circ C$, the Fe/Al/Cu catalyst separated into three reduction temperatures. The first reduction temperature was caused by the reduction of CuO species, the second by a reduction of $Fe_2O_3$ ($Fe^{3+}$) to $Fe_3O_4$ ($Fe^{8/3+}$), and the third by a reduction of $Fe_3O_4$ ($Fe^{8/3+}$) to FeO ($Fe^{2+}$). $Fe_2O_3$ ($Fe^{3+}$), the second reduction temperature, was reduced to $Fe_3O_4$ ($Fe^{8/3+}$), and the $Fe_3O_4$ ($Fe^{8/3+}$) generated was an active species in WGS. The Fe/Al/Cu

catalyst's second reduction temperature began at a lower temperature than the other catalysts. This means that the Fe/Al/Cu catalyst had an easy reducibility due to the synergistic effect between Fe/Al and Cu. **[FACP]** Similar to the aforementioned findings, the FAC-CP catalyst exhibited the first reduction temperature as reduced CuO species, the second reduction temperature as reduction of $Fe_2O_3$ to $Fe_3O_4$, and the third reduction temperature as reduction of $Fe_3O_4$ ($Fe^{8/3+}$) to FeO ($Fe^{2+}$). The FAC-CP catalyst exhibited the easier reducibility because it had the lowest reduction temperature for the $Fe_2O_3$ species. **[FACS]** Bare Fe showed two reduction temperatures of 405 and 700 °C, where $Fe_2O_3$ is reduced to $Fe_3O_4$ and $Fe_3O_4$ to FeO, respectively [28]. The reduction temperatures of Fe species in the scaled-up FAC catalyst shifted to a lower temperature compared to bare Fe. This suggests that the presence of Al and Cu facilitates the formation of the $Fe_3O_4$ active phase. Reduction temperatures for FAC-PC-1 and FAC-PC-3 were detected at substantially lower temperatures than for the other catalysts. As a consequence, both FAC-PC-1 and FAC-PC-3 were predicted to exhibit outstanding catalytic activity with enhanced reducibility in the HT-WGS process [28].

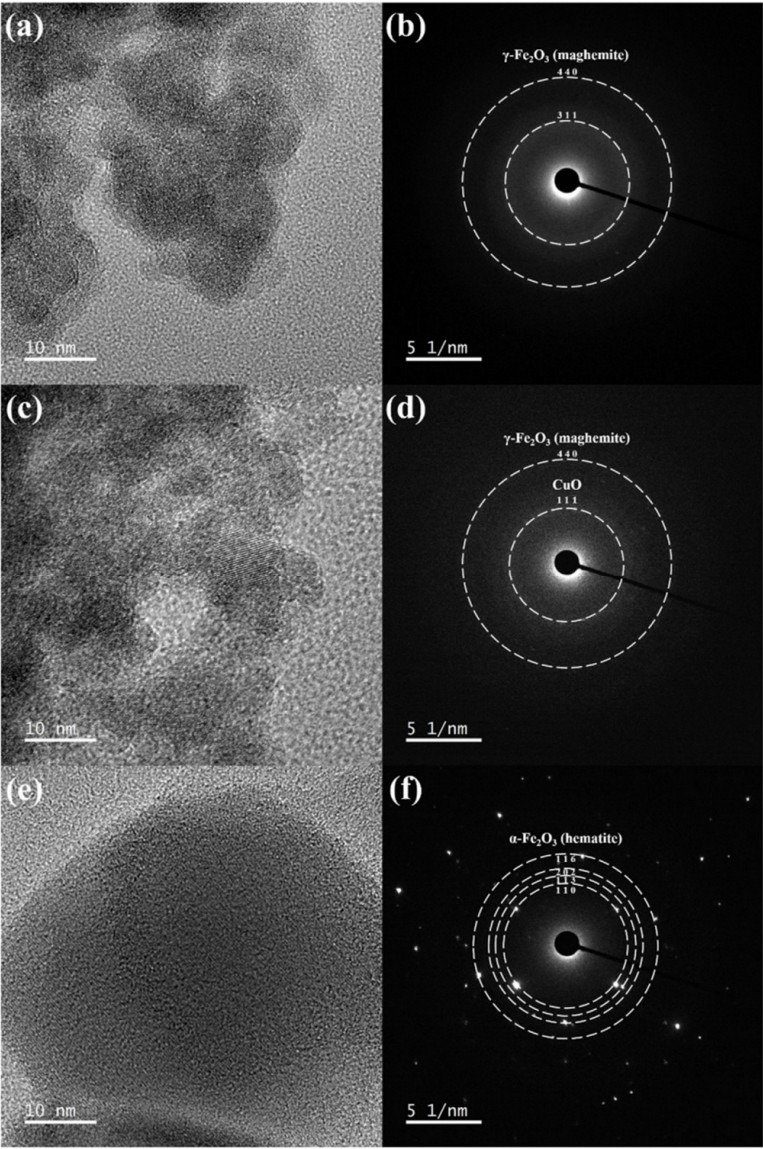

**Figure 1.** TEM images and SAED patterns of fresh FAC-PC-3, FAC-PC-3-240, and Fe-Al catalysts: (**a**,**b**) FAC-PC-3, (**c**,**d**) FAC-PC-3-240, (**e**,**f**) SAED Fe-Al. Adapted from Ref. [28]. Copyright 2019 Elsevier.

**Table 3.** Reduction characteristics of Fe-based catalysts.

| Catalyst | CuO to Cu$^0$ (°C) | Fe$_2$O$_3$ to Fe$_3$O$_4$ (°C) | Fe$_3$O$_4$ to FeO (°C) | Ref. |
|---|---|---|---|---|
| Fe/Al | - | 381 | 580 | |
| Fe/Al/Cu | 178 | 325 | 660 | [12] |
| Fe/Al/Ni | - | 380 | 580 (560 Ni) | |
| FAC-CP | 154 | 177 | 600~700 | |
| FAC-SG | 166 | 203 | 600~700 | [11] |
| FAC-IM | 178 | 325 | 660 | |
| Bare Fe | - | 405 | 700 | [28] |
| FAC-PC-1 | 126 | 155 | - | |
| FAC-PC-3 | 127 | 159 | - | |
| FAC-PC-5 | 153 | 185 | - | [28] |
| FAC-PC-10 | 205 | 253 | - | |
| FAC-PC-3-240 | 128 | 158 | - | |

**[FACP]** XPS analysis was performed to investigate the active species of Fe/Al/Cu catalysts synthesized using different production processes, and the results are outlined in Figure 2. As a result of XPS analysis, the Fe 2p peaks showed Fe 2p$_{1/2}$ and Fe 2p$_{3/2}$ peaks due to spin-orbit coupling, while the FAC-CP and FAC-IM catalysts showed similar tendencies. The peak at 710.8 eV was caused by Fe$_2$O$_3$. The FAC-SG catalyst changed slightly to 708.5 eV to suggest the Fe$_3$O$_4$ phase, which matches the XRD data. Cu 2p was deconvolved into three peaks, which are classified as reduced Cu species (Cu$^{1+}$/Cu$^0$), CuO (Cu$^{2+}$), and satellite peaks generated by CuO from Cu$^{2+}$. The surface composition of the catalyst was determined by calculating the area of the peak, and 47% of reduced Cu species were found on the surface of the FAC-CP catalyst. This indicates that the concentration of reduced Cu species was greater than that of other catalysts. The greater the concentration of the reduced Cu species, the greater the activity in the WGS reaction; hence, the FAC-CP catalyst is predicted to have more activity in the WGS process [11]. The catalytic reaction results are compatible with the results of the catalyst properties analysis and are presented in Figure 3. These data suggest that reducibility has a significant effect on the CO conversion rate. **[FAC]** Based on the calculated deconvoluted peak areas in Figure 2b, it was confirmed that the FAC-CP catalyst had 47% of reduced Cu species. This indicates that the concentration of reduced Cu species was greater than that of other catalysts. The greater the concentration of the reduced Cu species, the greater the activity in the WGS reaction; hence, the FAC-CP catalyst is predicted to have more activity in the WGS process [11]. The catalytic reaction results are compatible with the results of the catalyst properties analysis and are presented in Figure 3. These data suggest that reducibility has a significant effect on the CO conversion rate. **[FAC]** When the Fe/Al and metal-oxide-doped catalysts were reacted at a GHSV of 40,057 h$^{-1}$ and the results were compared, the Fe/Al/Cu catalyst showed the highest CO conversion rate across all temperature ranges. The Fe/Al/Cu catalyst demonstrated high CO conversion rates of 74.0% at 350 °C, 84.0% at 400 °C, 80.9% at 450 °C, and 76.9% at 500 °C. Over the reaction temperature range, the Fe/Al/Ni catalyst demonstrated rather stable CO conversion rates, but lower CO conversion than the Fe/Al/Cu catalyst. The following were derived from the CO conversion rate results. (1) The reducibility of catalyst is an essential factor in the HTS reaction, as shown by TPR findings. As a result of TPR, the Fe/Al/Cu catalyst had the lowest reduction temperature, indicating easy reducibility when compared to other catalysts, and the CO conversion rate indicated that it had superior catalytic performance than other catalysts. (2) Because of the synergistic effect of Cu, it provides a new active site and improves catalytic performance in the HTS process. Cu metal provides active oxygen species to oxidize CO to CO$_2$ by temporarily reducing CuO to Cu, and Cu is oxidized again to obtain oxygen from H$_2$O in HT-WGS [12]. Furthermore, the Fe/Al/Cu catalyst exhibited 100% CO$_2$ selectivity and 0% CH$_4$ selectivity, as well as very stable CO conversion after a 100 h stability test at 400 °C [12]. **[FACP]** Catalytic activity analysis of the Fe/Al/Cu

catalyst for each production process was also performed under the identical conditions as the previous research findings (GHSV = 40,057 h$^{-1}$, CO concentration = 38.2%). The FAC-CP catalyst was projected to have increased catalytic activity owing to its easier reducibility, according to the findings of catalyst characterization. The FAC-CP catalyst demonstrated the maximum CO conversion in all temperature ranges from 350 to 550 °C, as predicted. This is related to the reducibility of the catalyst, which is consistent with earlier study findings. As implied by TPR, the FAC-CP catalyst has the lowest reduction temperature, and redox reaction from $Fe^{2+}$ to $Fe^{3+}$ happens. Furthermore, the catalytic activity is influenced by the easy reducibility of CuO species, the high BET surface area, and the small crystal size. The FAC-CP catalyst demonstrated the maximum thermal stability without catalytic deactivation after 25 h of time on stream at 450 °C under a GHSV of 40,057 h$^{-1}$ [11]. **[FACS]** Finally, the reaction of the scaled-up FAC catalysts resulted in CO conversion rates of 94.9 and 95.4% at 350 °C for FAC-PC-1 and FAC-PC-3, respectively. They performed well in all temperature ranges with 100% $CO_2$ selectivity and 0% $CH_4$ selectivity, and the process proceeded without any side reactions. This is due, like the previous studies, to the high BET surface area, small $Fe_3O_4$ crystal size, great reducibility, and high degree of Cu dispersion [28]. According to the previous study's findings, substituting the Fe/Cr catalyst with a Fe-/Al-based catalyst enhances catalytic stability in the HT-WGS process. Furthermore, since Al (0.675 Å) and Fe (0.690 Å) have comparable ionic radii, Al may be readily absorbed into the Fe lattice, which can substitute Cr-free catalysts.

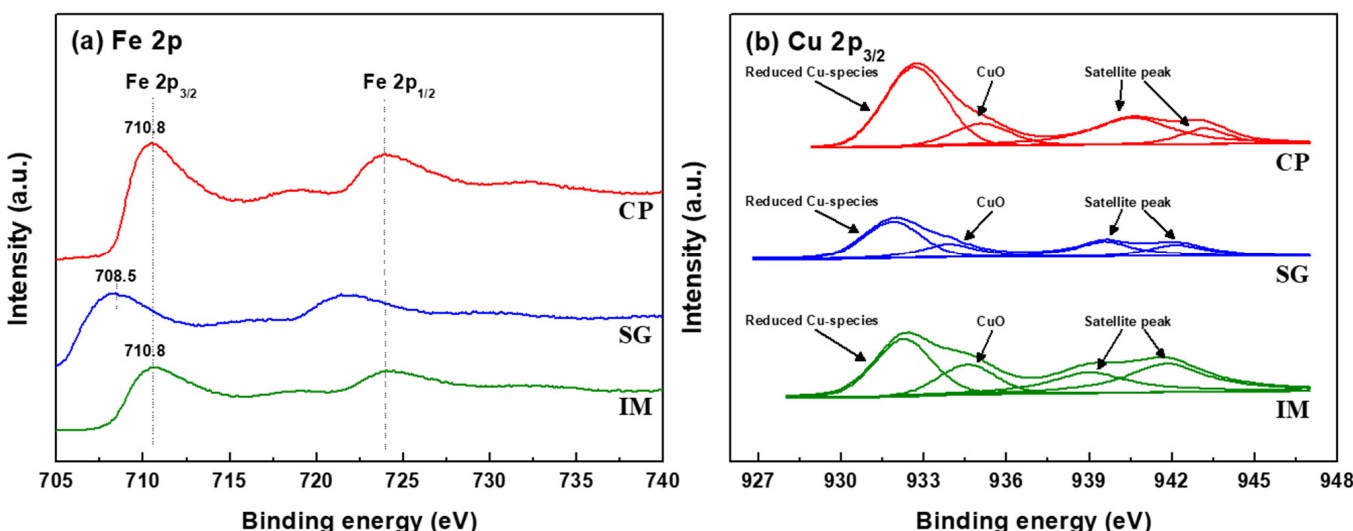

**Figure 2.** XPS spectra of the fresh FAC catalysts prepared by different methods. (**a**) Fe 2p; (**b**) Cu 2p. Adapted from Ref. [11]. Copyright 2015 Elsevier.

**[CFMA]** In a previous study, a $CuFe_2O_4$ catalyst integrated with mesoporous alumina was investigated, and it was proven that at low temperatures, $Fe_2O_3$ was reduced to $Fe_3O_4$ and had better catalytic activity. Meanwhile, Cu enhances reduction and speeds the WGS process, replaces $Fe^{2+}$ with $Cu^{2+}$, increases electron hopping between $Fe^{2+}$ and $Fe^{3+}$, and improves catalytic activity via the creation of $Fe^{2+}/Fe^{3+}$ and $Cu^{2+}/Cu^{+}$ redox pairs. The use of additional metals, such as Ni and Co, in addition to Cu, was shown to be advantageous in boosting the CO conversion rate; thus, the change in catalytic activity was investigated. The HTS process was catalyzed by mesoporous alumina (MA) and spinel ferrite ($MFe_2O_4$) catalysts (M = Ni, Co, Fe, Cu). Table 4 summarizes the physical characteristics of catalysts. The $CoFe_2O_4$-MA catalyst has a $Co_3O_4$ spinel structure with the lowest crystallite size (3.7 nm). When Co, Ni, and Cu are doped here, the oxidation state of Fe shifts from $Fe^{2+}$ to $Fe^{3+}$. The Cu-doped catalyst switched from metal ferrites ($MFe_2O_4$) to magnetite ($Fe_3O_4$), which is a reduction temperature, at a lower temperature as a consequence of TPR to prove the reducibility of the catalyst. This is due to the fact that $CuFe_2O_4$ is quickly transformed

to Cu and $Fe_3O_4$, allowing for easier reduction than other catalysts. The $CuFe_2O_4$-MA catalyst had the largest CO conversion rate of 69% at 350 °C as a result of the process. The reason for this is that copper ferrite is better converted to magnetite at lower temperatures. Furthermore, all of the catalysts displayed 100% $CO_2$ selectivity and 0% $CH_4$ selectivity. The $CuFe_2O_4$-MA catalyst utilizing Cu demonstrated the largest catalytic performance among different metal materials, optimizing the Cu/Fe ratio. The $Cu_xFe_{(3-x)}O_4$-MA catalyst was synthesized and examined to compare the structure and physicochemical characteristics according to the Cu/Fe ratio. Catalysts were produced using the SG method in various Cu/Fe ratios of 0.2, 0.5, 1, 2, and 5, which were designated as $Cu_{0.5}Fe_{2.5}O_4$-MA (CFMA-5), $Cu_{1.0}Fe_{2.0}O_4$-MA (CFMA-10), $Cu_{1.5}Fe_{1.5}O_4$-MA (CFMA-15), $Cu_{2.0}Fe_{1.0}O_4$-MA (CFMA-20), and $Cu_{2.5}Fe_{0.5}O_4$-MA (CFMA-25), respectively. The catalyst's crystallite size increased as the Cu/Fe ratio rose. However, following the reaction, CFMA-20 and CFMA-25 catalysts were found to have greater crystal sizes than other catalysts. Table 4 summarizes these findings. As the Cu/Fe ratio rose, the BET surface area and pore volume decreased. The $Fe^{2+}/Fe^{3+}$ ratios were calculated as 1.3 (CFMA-5 cat.), 1 (CFMA-10 cat.), 0.7 (CFMA-15 cat.), 1.4 (CFMA-20 cat.), and 1.9 (CFMA-25 cat.), and the CFMA-15 catalyst had the highest $Fe^{3+}$ ratio among them. The CFMA-15 catalyst has the largest $Fe^{3+}$ concentration based on area calculations. This is due to the abundance of $CuFe_2O_4$ formation on the surface. All catalysts produced in Cu 2p were found to have similar positions. As a result of estimating the catalyst's surface composition, the concentration of reduced Cu species grew until the Cu/Fe ratio reached 1, at which point bulk CuO species formed on the surface. All catalysts revealed just one reduction temperature between 120 and 300 °C as a result of TPR to validate the reducibility of the catalyst. This reduction temperature appears when reduced CuO and $CuFe_2O_4$ are converted to CuO and $Fe_2O_3$. At the lowest temperature of 179 °C, the CFMA-15 catalyst reduced CuO species. This is due to the fact that Cu species ($Cu^+$, $Cu^0$) are quickly reduced on the surface. $CuFe_2O_4$ is reduced to Cu and $Fe_3O_4$ using the reduced Cu species ($Cu^+$) as a hydrogen donor center. The displacement of the CuO reduction temperature boosts the reduction capacity of catalyst and demonstrates significant activity in HTS due to the strong interaction between Cu/Fe and mesoporous alumina. Furthermore, the CFMA-15 catalyst shows the largest $H_2$ consumption and is projected to be very active in the HT-WGS process. Figure 3 shows the catalytic reaction results. The CFMA-15 catalyst had the largest CO conversion rate, as predicted by the above-mentioned properties. Furthermore, all catalysts demonstrated 100% $CO_2$ selectivity and 0% $CH_4$ selectivity. The spinel copper and tetragonal structures were observed in the fresh and utilized CFMA-15 catalysts. This improves catalytic activity by increasing electron hopping between $Fe^{2+}$ and $Fe^{3+}$. The CFMA-15 catalyst demonstrated a CO conversion rate of 78% at 450 °C and sustained catalytic activity throughout time on stream. As a result, the CFMA-15 catalyst with optimized Cu/Fe ratio seems to have better activity in WGS than the previous study's CFMA-10 catalyst [13]. As a result of conducting the reaction with Fe-based catalysts under the GHSV of about 40,000~42,000 $h^{-1}$, the FAC-PC-3 catalyst showed a highest CO conversion in the temperature range of 350 to 450 °C. Related to the reaction results, the BET surface area, $Fe_3O_4$ crystallite size, reducibility, and Cu dispersion of all Fe-based catalysts depended on precursor concentration. The high activity of the FAC-CP-3 catalyst is attributed to the easy reducibility of $Fe_2O_3$ and Cu active species according to the characterization results. Therefore, the FAC catalyst was developed enough to replace the commercial Fe-Cr catalyst and showed commercialization potential.

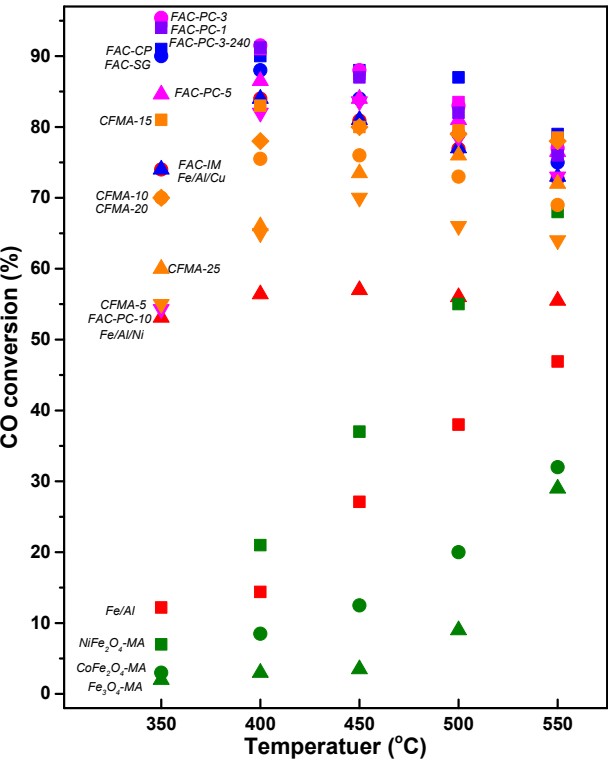

**Figure 3.** CO conversion of Fe-based catalysts.

**Table 4.** Characteristics of $Fe_3O_4$-MA and $MFe_2O_4$-MA (M=Cu, Ni, Co) and $Cu_xFe_{(3-x)}O_4$-MA (x = 0.5, 1.0, 1.5, 2.0, 2.5) catalysts.

| Catalyst | BET Surface Area ($m^2/g$) | | Crystallite Size (nm) | | Crystallite Size of Metallic Cu (nm) [b] |
|---|---|---|---|---|---|
| | Fresh | Used | Fresh [a] | Used [a] | |
| $CuFe_2O_4$-MA | 163 | 110 | 7.8 | - | - |
| $NiFe_2O_4$-MA | 212 | 107 | 7.5 | - | - |
| $CoFe_2O_4$-MA | 188 | 59 | 3.7 | - | - |
| $Fe_3O_4$-MA | 176 | 91 | 10.6 | - | - |
| CFMA-5 | 176 | 82 | 8.0 | 12.6 | N.A. [c] |
| CFMA-10 | 163 | 110 | 7.8 | 12.3 | 18.2 |
| CFMA-15 | 97 | 72 | 9.4 | 12.2 | 19.1 |
| CFMA-20 | 42 | 41 | 11.3 | 13.8 | 23.7 |
| CFMA-25 | 36 | 23 | 17.3 | 19.6 | 26.1 |

[a] Calculated from (3 1 1) peak of spinel ferrite using the Scherrer equation. [b] Calculated from (2 0 0) peak of metallic Cu using the Scherrer equation. [c] Not available due to very broad and weak XRD peaks.

### 3.2. Cu-Based Catalysts (Excellent Redox Ability of Cu)

Because of the toxicity of Cr, a commercial catalyst used in the HTS reaction, studies on Cr-free catalysts have been conducted for over 20 years. Among them, Cu-based catalysts are now being researched. Cu-based catalysts are less expensive than precious metals and exhibit high activity in the WGS process, making them popular as active metals in catalysts. Numerous studies are being conducted to introduce various supports into Cu [90]. Because of its low catalytic activity and stability, $CeO_2$ is not employed as an active metal in the catalyst. When $CeO_2$ is utilized as a support, it offers a larger surface area to boost metal oxide dispersion and increases water dissociation in the WGS process [91,92]. In this section, three papers related to Cu-based catalysts are summarized, and descriptions are distinguished by naming [NCC], [CCA], and [CBCA] in front of descriptions for each characteristic analysis.

**[NCC]** A recent study found that the mesoporous Ni-Cu-CeO$_2$ catalyst (NCC) had higher activity than a single metal oxide catalyst in the HTS process. The developed and optimized Ni-Cu-CeO$_2$ catalyst outperformed the single metal oxide catalysts in terms of performance, and the methanation reaction, a side reaction, did not occur. The combination of CeO$_2$ with Ni and Cu metal oxides was shown to have greater activity and thermal stability than the previous Cu-CeO$_2$ catalyst. Various synthesis methods were applied to improve the performance of the Ni-Cu-CeO$_2$ catalyst. The Ni-Cu-CeO$_2$ catalysts were made via evaporation-induced self-assembly (EISA), CP, solvothermal (ST) method, and IM, and were designated Ni-Cu-CeO$_2$-SG (NCC-SG), Ni-Cu-CeO$_2$-CP (NCC-CP), Ni-Cu-CeO$_2$-ST (NCC-ST), and Ni-Cu-CeO$_2$-IM (NCC-IM), respectively. Table 5 outlines the physical parameters of Ni-Cu-CeO$_2$ catalysts synthesized using different approaches. The NCC-SG catalyst generated by the EISA method had the lowest crystal size (5 nm) and the highest surface area (102 m$^2$/g) of the developed catalysts. Table 6 depicts the TPR data used to validate the catalyst's reducibility. All of the produced catalysts displayed three reduction temperatures in the temperature range of 100 to 400 °C. The highest reduction temperature (187 °C) for CuO interacting with the support was observed in NCC-CP, which might be attributable to the almost unreduced Cu species trapped in the CeO$_2$ lattice. Because the TPR findings of the EISA and CP catalysts were comparable, in the HTS process, oxygen vacancy is known to inhibit sintering by stabilizing the transition metal nanoparticles supported on the oxide surface. Figure 4 depicts the catalytic reaction results. Among the produced catalysts, EISA and CP demonstrated comparable CO conversion rates in the temperature range of 350 to 550 °C. The EISA method's high activity is attributed to the mesoporous architecture scattered in active spots on the catalyst surface. The interaction between the CeO$_2$ support and the metal oxide to generate a form in which Cu and Ni are reduced on the catalyst surface is responsible for the high activity of the NCC-CP catalyst. To compare the activities of the two catalysts in more detail, the activity was assessed at a higher GHSV of 161,000 h$^{-1}$, and the catalyst produced using the EISA method had greater activity. This is due to the mesoporous structure, which enables gas to move freely while maintaining high activity. Furthermore, as a consequence of EISA, the catalyst synthesized by the EISA method has a high reaction rate in all temperature ranges. This finding showed that as the surface area increased, so did the response rate. The phase shift of the catalyst after the reaction. Because Ni was equally diffused in the catalyst, it did not reveal the oxide phase of Ni in the catalysts produced using the EISA and CP methods. The stability test of the EISA and CP catalysts revealed that both demonstrated stable performance when reacting at 450 °C for 25 h, with the EISA method catalyst showing somewhat greater activity. Thus, it was determined that this is an appropriate catalyst for the HTS reaction [26]. At high temperatures, the CeO$_2$ catalyst initiates the WGS process through a redox mechanism. The redox process adsorbs CO on the metal site and subsequently oxidizes it with CeO$_2$ lattice oxygen to produce CO$_2$ and oxygen vacancy. H$_2$O oxidizes reduced CeO$_2$ again to create hydrogen. According to Djinovic et al., the activity of Cu-based catalysts employing CeO$_2$ as a support in the WGS process is owing to the strong interaction between CuO and CeO$_2$, which influence the creation of oxygen vacancy. Previous research has shown that oxygen vacancy is directly related to catalytic performance in WGS reactions. **[CCA]** The activity difference between several synthesis techniques of Cu/$\gamma$-Al$_2$O$_3$ (CuA) catalyst utilizing CeO$_2$ as a support was investigated in this study. Ce/Cu/$\gamma$-Al$_2$O$_3$ (CeCuA) catalysts, Cu/Ce/$\gamma$-Al$_2$O$_3$ (CuCeA) catalysts, and Cu-Ce/$\gamma$-Al$_2$O$_3$ (Cu-CeA) catalysts were developed based on the impregnation order difference from the Cu/$\gamma$-Al$_2$O$_3$ catalyst. The catalyst surface area declined in the following order: CuA (124.5 m$^2$/g) > CuCeA (116.1 m$^2$/g) > Cu-CeA (114.0 m$^2$/g). Table 5 summarizes the N$_2$O chemisorption data. The CeCuA catalyst was found to have a high Cu dispersion of 1.7% and a small Cu crystal size. The TPR result to validate the catalyst reduction is given in Table 6. At a lower temperature than the other catalysts, the CeCuA catalyst deconvolved into three reduction temperatures. Accordingly, the CeCuA catalyst should be quite active

in the WGS process. These findings indicate that adding CeO$_2$ to the CuA catalyst surface improves the redox characteristics of copper oxide.

**Table 5.** Characteristics of Cu-based catalysts.

| Catalyst | BET Surface Area (m²/g) [a] | Cu Surface Area (m²/g) [b] | Crystallite Size (nm) | | | | | Lattice Parameter | Cu Dispersion (%) [b] | Ref. |
|---|---|---|---|---|---|---|---|---|---|---|
| | | | Fresh [a] | Used [a] | Cu [b] | CuO [c] | CeO$_2$ [c] | | | |
| CeO$_2$ | 81 | - | 9 | - | - | - | - | 5.45 | - | [26] |
| NCC-SG | 102 | - | 5 | 6 | - | - | - | 5.43 | - | [26] |
| NCC-CP | 96 | - | 8 | 8 | - | - | - | 5.39 | - | [26] |
| NCC-ST | 79 | - | 10 | 11 | - | - | - | 5.41 | - | [26] |
| NCC-IM | 14 | - | 13 | 13 | - | - | - | 5.34 | - | [26] |
| CuA | 163.4 | 2.5 | - | - | 34.4 | - | - | - | 2.9 | [27] |
| CeCuA | 124.5 | 2.2 | - | - | 50.7 | 27.4 | 9.6 | - | 1.7 | [27,48] |
| Cu-CeA | 114.0 | 2.0 | - | - | 56.3 | - | - | - | 1.5 | [27] |
| CuCeA | 116.1 | 1.5 | - | - | 75.4 | - | - | - | 1.2 | [27] |
| CBCA | 114.0 | - | - | - | - | 28.2 | 9.8 | - | 1.7 | [48] |
| CZCA | 111.9 | - | - | - | - | 29.4 | 10.2 | - | 1.5 | [48] |
| CNCA | 122.0 | - | - | - | - | 30.2 | 9.9 | - | 1.4 | [48] |

[a] Estimated from N$_2$ adsorption isotherm at −196 °C. [b] Estimated from N$_2$O chemisorption. [c] Calculated from XRPD.

**Table 6.** Reduction Temperature of Cu-based catalysts.

| Catalyst | CuO Species Interacted with CeO$_2$ (°C) | CuO Species Not Interacted with CeO$_2$ (°C) | Bulk CuO (°C) | Ref. |
|---|---|---|---|---|
| NCC-SG | 169 | 257 | - | |
| NCC-CP | 187 | - | - | [26] |
| NCC-ST | 165 | 200 | 265 | |
| NCC-IM | 182 | 261 | 302 | |
| CuA | - | - | 197 | |
| CeCuA | 136 | 158 | 183 | [27] |
| Cu-CeA | 141 | - | 177 | |
| CuCeA | - | - | 188 | |
| CBCA | 170 | 186 | 215 | |
| CZCA | 134 | 163 | 203 | [48] |
| CNCA | 114 | 143 | 180 | |

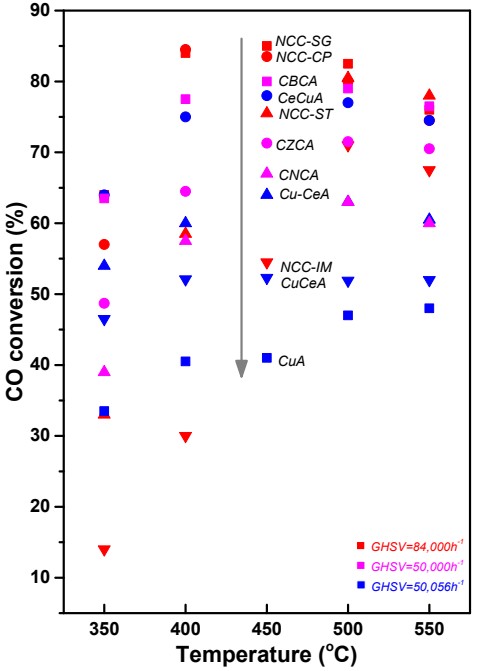

**Figure 4.** CO conversion of Cu-based catalysts.

Figure 5 shows the Raman spectroscopy data. In addition, it has the highest A600/A460 value when compared to other catalysts. Figure 4 shows the reaction results used to determine the activity of the catalyst. In comparison to other catalysts, the CeCuA catalyst had the highest CO conversion rate throughout the process. This is consistent with earlier analysis and oxygen vacancy results. All catalysts had 0% and 100% $CH_4$ and $CO_2$ selectivity, respectively. The CeCuA catalyst has the largest turn over frequency (TOF) value, which explains why it has the maximum activity in HTS. In addition, the $E_a$ value was $67 \pm 1$ kJ/mol, which was the lowest when compared to other catalysts. The CeCuA catalyst slightly decreased from 78% to 76% in the first 5 h when the CO conversion rate was reacted with a GHSV of 50,056 $h^{-1}$ at 450 °C as a result of time on stream to evaluate the stability of the catalyst, but stable performance was maintained for 40 h. According to the redox mechanism of the WGS reaction, the concentration of oxygen vacancy is a significant component in the WGS reaction, as is the amount of reduced Cu species. In the WGS process, the reduced copper species are catalytically active. Consequently, the CeCuA catalyst demonstrated the best CO conversion despite a very high GHSV of 50,056 $h^{-1}$ and a high CO concentration in the reactant gas [27]. Because of the high concentration of oxygen vacancy and the substantial amount of reduced Cu species, the CeCuA catalyst demonstrated strong catalytic activity in the HTS process. However, significant enhancements to the performance of current CeCuA catalysts are necessary to boost the efficiency of waste-to-hydrogen production. **[CBCA]** Ba, Zr, and Nd were doped into Ce/Cu/$\gamma$-$Al_2O_3$ catalysts in this study to examine their influence on the physicochemical parameters and catalytic performance of HTS. Table 5 summarizes the catalyst's physical properties. As a result of BET surface area analysis, it decreased in the order of Ce/Cu/$\gamma$-$Al_2O_3$ (CeCuA) > Ce-Nd/Cu/$Al_2O_3$ (CNCA) > Ce-Ba/Cu/$Al_2O_3$ (CBCA) > Ce-Zr/Cu/$Al_2O_3$ (CZCA). The copper oxide crystal size of the CeCuA catalyst is 27.4 nm; whereas, the copper oxide crystal size of the Ba-doped catalyst is 28.2 nm. The CeCuA catalyst has the maximum Cu dispersion of 1.7%, followed by the Zr-doped catalyst at 1.5%, and the Ba-doped catalyst at 1.7%. The Ba-doped catalyst has a comparable Cu dispersion to conventional catalysts and is predicted to have stronger catalytic activity than other catalysts. Figure 5 shows the Raman spectrum results. All catalysts had a maximum between 460 and 600 $cm^{-1}$. The oxygen vacancy in the CBCA and CZCA catalysts was high. Table 6 shows the reduction characteristics of prepared catalysts. The CBCA and CZCA catalysts moved to a higher temperature as a consequence of the analysis than the CeCuA catalysts. Figure 4 shows the strong metal to support interaction (SMSI) results. Because the SMSI of CBCA and CZCA catalysts may block copper sintering, these catalysts have greater catalysts during WGS reaction than CeCuA and CNCA catalysts. It is anticipated to show stable catalytic activity. The CBCA catalyst demonstrated better catalytic activity than other catalysts as a consequence of the catalytic process. This is because the addition of Ba increased the concentration of oxygen vacancy. The TOF decreased in the following order: CBCA > CZCA > CeCuA > CNCA. The CBCA and CZCA catalysts demonstrated greater catalytic stability than other catalysts and were reacted at 500 °C for 40 h at 50,000 $h^{-1}$ GHSV. The lowest hydrogen yield reduction rate (27.9%) and production rate (36.1%) were achieved by the CBCA catalyst. This is mostly owing to its high resistance to sintering by Ba addition and its complete $CO_2$ selectivity. The addition of Ba and Zr to the CeCuA catalyst enhanced the HTS reaction activity and SMSI effect, which raised the oxygen vacancy concentration and improved the catalyst stability. Nd doping, on the other hand, enhances the reducibility of the catalyst but has low catalytic efficiency. The association between characterization data and catalytic performance was found to be highly reliant on the interaction between the oxygen vacancy concentration supporting the interaction and the metal supported on the supporter. Because the SMSI effect was enhanced and the oxygen vacancy concentration was raised, the CBCA catalyst in the produced CeCuA demonstrated outstanding catalytic performance for HTS reaction. Therefore, the CBCA catalyst is the best choice for the HTS reaction to boost hydrogen production efficiency from waste-simulating syngas [48]. As a result of the reaction of the NCC catalyst, the NCC-SG catalyst showed high catalytic

activity at a GHSV of 84,000 h$^{-1}$. The higher activity of the NCC-SG catalyst was due to the mesoporous nature of the catalyst which provides the higher surface area and facilitates the uninterrupted diffusion of molecules to and from active sites of the catalysts. Other Cu-based catalysts reacted at GHSV of 50,000 h$^{-1}$; CBCA catalysts showed high catalytic activity at 400~550 °C. The higher activity of the CBCA catalyst was due to enhanced SMSI effect and increased oxygen vacancy concentration.

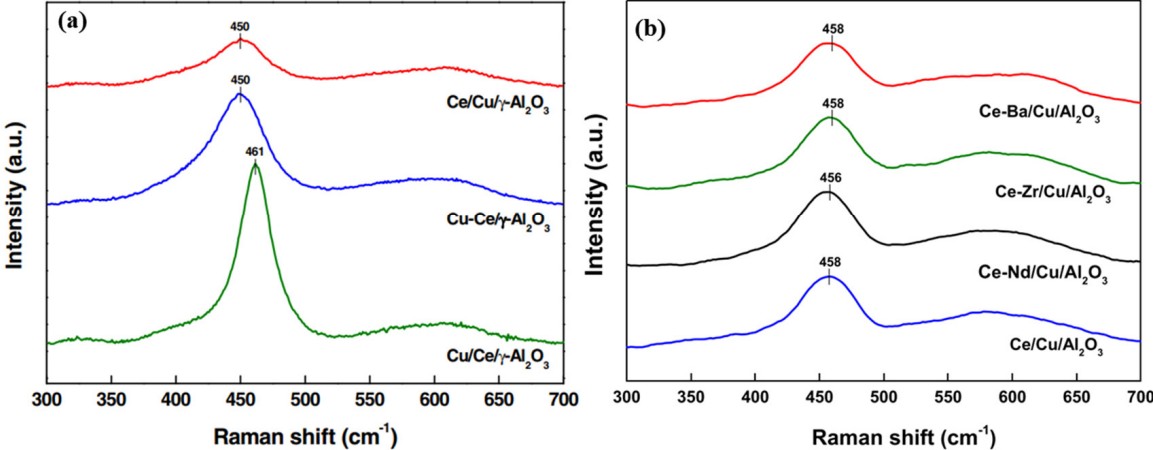

**Figure 5.** Raman spectra of the Cu-based catalysts: (**a**) Cu/γ–Al$_2$O$_3$ catalyst and CeO$_2$–promoted Cu/γ–Al$_2$O$_3$ catalysts. Adapted from Ref. [27]. Copyright 2016 Elsevier. (**b**) Ce/Cu/Al$_2$O$_3$ catalysts with various additives. Adapted from Ref. [48]. Copyright 2020 Elsevier.

### 3.3. Ni-Based Catalyst (High OSC)

A prior study showed outstanding catalytic activity by doping a CeO$_2$ support with metal oxides, such as Ni and Cu, in order to develop a Cr-free catalyst for the HT-WGS process. CeO$_2$ reacts significantly with active metal compounds, resulting in increased catalytic characteristics. Among numerous non-precious metals, Ni exhibits outstanding activity by increasing catalytic surface area [93,94]. Thus, it is a suitable catalyst for the HT-WGS process. However, since methanation reaction is a side reaction of Ni-based catalysts, research on improving the activity of the catalyst without methanation reaction is necessary. According to research, alkali metals impede the methanation process. In this section, one paper related to Ni-based catalysts is summarized. Previous research found that Fe-Al-Ni catalysts doped with Ba helped decrease side reactions. The Ni-CeO$_2$ catalyst was doped with the alkali and alkaline promoters K, Ca, Mg, and Ba. This catalyst was chosen to inhibit a key side reaction, methanation, and the activity and side reaction of the produced catalyst, which were investigated. Table 7 summarizes the physical parameters of Ni-based catalysts. The Ni-CeO$_2$ catalyst has the largest BET surface area among the prepared catalysts, according to the BET analysis. The BET surface area of the promoter/Ni-CeO$_2$ catalyst decreased relative to the catalyst without the promoter when the promoter was introduced. The prepared catalysts had a BET surface area in the descending order of Ni-CeO$_2$ (131.29 m$^2$/g) > Ba/Ni-CeO$_2$ (118.83 m$^2$/g) > Ca/Ni-CeO$_2$ (112.45 m$^2$/g) > K/Ni-CeO$_2$ (110.34 m$^2$/g) > Mg/Ni-CeO$_2$ (109.85 m$^2$/g). The XRD data are given in Figure 6A, and peaks associated with CeO$_2$ were found at 28.6°, 33.1°, 47.5°, 56.4°, 59.1°, 69.4°, 76.7°, and 79.1°. This is the cubic phase of CeO$_2$ (ICDD card No. 34-0394). Peaks related to Ni were also discovered at 44.7° and 51.7°, which are cubic Ni$^0$ (ICDD card No. 98-6960). The promoter peak was not visible because of the low concentration of 2%. Table 7 lists the H$_2$ chemisorption data. The active metal dispersion decreased in the order of Ni-CeO$_2$ (4.00%) > Ca/Ni-CeO$_2$ (3.28%) > Ba/Ni-CeO$_2$ (3.20%) > K/Ni-CeO$_2$ (3.16%) > Mg/Ni-CeO$_2$ (1.39%). It was proven that the dispersion decreased when the promoter was added. The dispersion of the Mg/Ni-CeO$_2$ catalyst was substantially smaller than that of other catalysts. This is due to a significant contact between MgO and NiO, which decreases the production of active Ni species capable of reacting with hydrogen. Table 7 summarizes

the results of the active Ni site calculation. The Ni-CeO$_2$ catalyst had the highest value of $1.37 \times 10^{-6}$ mol/g$_{cat}$, followed by the Ba/Ni-CeO$_2$ catalyst at $1.14 \times 10^{-6}$ mol/g$_{cat}$. The TPR analysis results are given in Figure 6B, and all catalyst reduction peaks were deconvolved into four peaks. The initial peak ($\alpha$) was caused by a decrease in the amount of oxygen adsorbed on the surface. Ni ions entered the CeO$_2$ lattice and a few Ce$^{4+}$ ions were replaced in the Ni-CeO$_2$ catalyst. Oxygen vacancy formed because of the difference in ionic radii of Ni$^{2+}$ (0.81 Å) and Ce$^{4+}$ (0.97 Å), and the adsorbed oxygen species could be readily reduced.

**Table 7.** Characteristics of Ni-based catalysts.

| Catalyst | BET Surface Area (m$^2$/g) [a] | | Crystallite Size (nm) | | | Ni$^0$ Dispersion [c] (%) | Active Ni Site [c] (mol/g$_{cat}$) | Ref. |
|---|---|---|---|---|---|---|---|---|
| | Fresh | Used | Fresh [b] (Fe$_2$O$_3$) | Used [b] (Fe$_3$O$_4$) | Ni$^0$ [c] | | | |
| Fe/Al/Ni | 81.4 | 11.2 | 14.9 | 20.6 | - | - | - | [12] |
| Ni-CeO$_2$ | 131.29 | - | - | - | 21.03 | 4.00 | $1.37 \times 10^{-6}$ | [87] |
| K/Ni-CeO$_2$ | 110.34 | - | - | - | 26.71 | 3.16 | $1.08 \times 10^{-6}$ | [87] |
| Ca/Ni-CeO$_2$ | 112.45 | - | - | - | 25.61 | 3.28 | $1.13 \times 10^{-6}$ | [87] |
| Mg/Ni-CeO$_2$ | 109.85 | - | - | - | 61.13 | 1.39 | $3.16 \times 10^{-6}$ | [87] |
| Ba/Ni-CeO$_2$ | 118.83 | - | - | - | 26.33 | 3.20 | $1.14 \times 10^{-6}$ | [87] |

[a] Estimated from N$_2$ adsorption isotherm at −196 °C. [b] Calculated from XRD. [c] Estimated from the H$_2$ chemisorption.

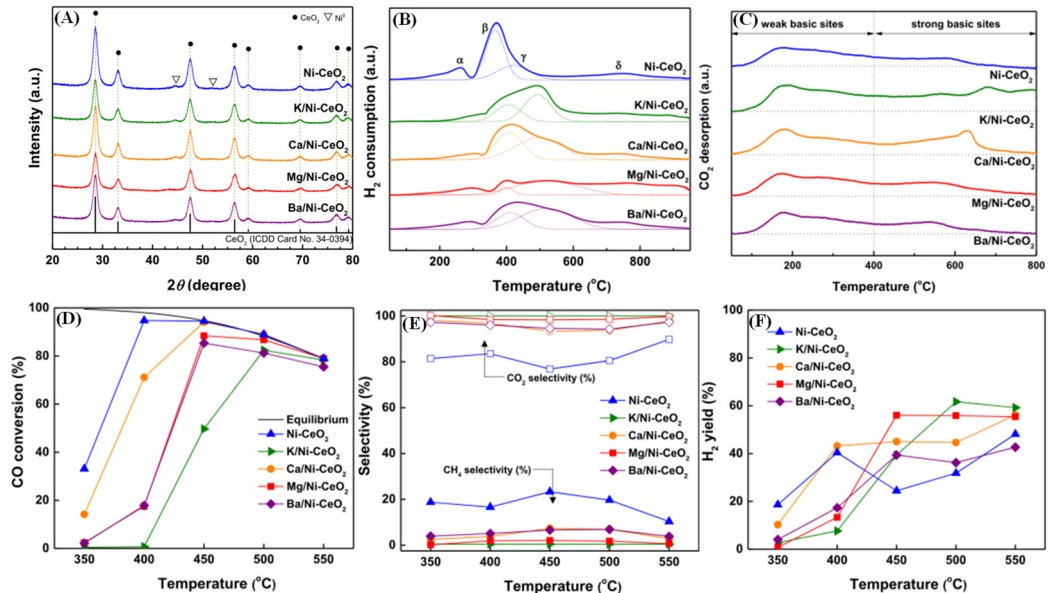

**Figure 6.** (**A**) XRD patterns of Ni-based catalysts; (**B**) H$_2$-TPR of Ni-based catalysts; (**C**) CO$_2$-TPD patterns of Ni-based catalysts; (**D**) CO conversion of Ni-based catalysts. (**E**) Selectivity to CH$_4$ and CO$_2$ of Ni-based catalysts; (**F**) H$_2$ yield of Ni-based catalysts. Adapted from Ref. [87]. Copyright 2022 Elsevier.

The second largest and broadest peak may be divided into two reduction peaks ($\beta$, $\gamma$), which correspond to free NiO and complex NiO reduction, respectively. Free NiO has a low interaction with CeO$_2$, but complex NiO has a high interaction with CeO$_2$, hindering the reduction. The last reduction peak ($\delta$) is generated by CeO$_2$ reduction. When compared to the traditional Ni-CeO$_2$ catalyst, the promoter-introduced catalyst is displaced to the right, showing an enhanced interaction between Ni and the support. It is well recognized that the catalyst's easy reducibility improves electron transfer capacity and consequently WGS activity. Thus, Ni-CeO$_2$ catalysts having NiO reduction peaks at low temperatures should have higher WGS activity. The CO$_2$-TPD results are displayed in Figure 6C, and the K/Ni-Ce O$_2$ catalyst exhibited the largest amount of CO$_2$ desorption. The prepared catalyst desorbs CO$_2$ at the following rates: K/Ni-CeO$_2$ (154.03 cm$^3_{CO2}$/g$_{cat}$) > Ca/Ni-CeO$_2$ (109.10 cm$^3_{CO2}$/g$_{cat}$) > Mg/Ni-CeO$_2$ (101.91 cm$^3_{CO2}$/g$_{cat}$) > Ba/Ni-CeO$_2$ (88.44 cm$^3_{CO2}$/g$_{cat}$)

> Ni-CeO$_2$ (87.75 cm$^3_{CO2}$/g$_{cat}$). It was proven that except for the K/Ni-CeO$_2$ catalyst, the methanation process was repressed and the WGS activity enhanced. The reaction results are given in Figures 6D and 7, and it was verified that all of the produced catalysts showed low activity at 350 °C. The CO conversion rates for the Ni/CeO$_2$ and Ca/Ni-CeO$_2$ catalysts were 33% and 14%, respectively; whereas, the other catalysts exhibited no activity. The Ni/CeO$_2$ and Ca/Ni-CeO$_2$ catalysts enhanced CO conversion rates to 94% and 71%, respectively, at 400 °C. The Ni/CeO$_2$ catalyst had the best CO conversion rate across all temperature ranges; whereas, the Ca/Ni-CeO$_2$ catalyst showed strong activity at low temperatures. This is due to the Ca/Ni-CeO$_2$ catalyst's surface having a high oxygen storage capacity (OSC). The Ni/CeO$_2$ catalyst, however, demonstrated that the methanation process happened in all temperature ranges, as shown in Figure 6E. When the promoter was introduced, the methanation reaction was repressed. Figure 6F shows the outcome of estimating the H$_2$ yield. The Ni-CeO$_2$ catalyst had the best yield at around 350 °C, but the Ca/Ni-CeO$_2$ catalyst grew fast at 400 °C. Mg/Ni-CeO$_2$ catalysts at 450 °C and K/Ni-Ce O$_2$ catalysts at 550 °C had the largest values, and Ca/Ni-Ce O$_2$ and Mg/Ni-CeO$_2$ catalysts also had large values. The Ca/Ni-CeO$_2$ and Mg/Ni-CeO$_2$ catalysts were reacted for 60 h at 450 °C with a GHSV of 315,282 h$^{-1}$ for the stability test results. Without deactivation, the Ca/Ni-CeO$_2$ and Mg/Ni-CeO$_2$ catalysts converted CO at rates of 91% and 69%, respectively. As a result, when compared to commercial and other catalysts, the Ca/Ni-CeO$_2$ catalyst had good hydrogen production capability as well as high activity and stability [87]. As a result of the reaction of the Ni-based catalyst, the Ca/Ni-CeO$_2$ catalyst exhibited superior HTS activity, compared to other promoted Ni-CeO$_2$ catalysts. The higher activity of Ca/Ni-CeO$_2$ catalyst was due to the high OSC. Similarly, the Ca/Ni-CeO$_2$ catalyst had stability for 18 h at a very high GHSV of 1,050,957 h$^{-1}$.

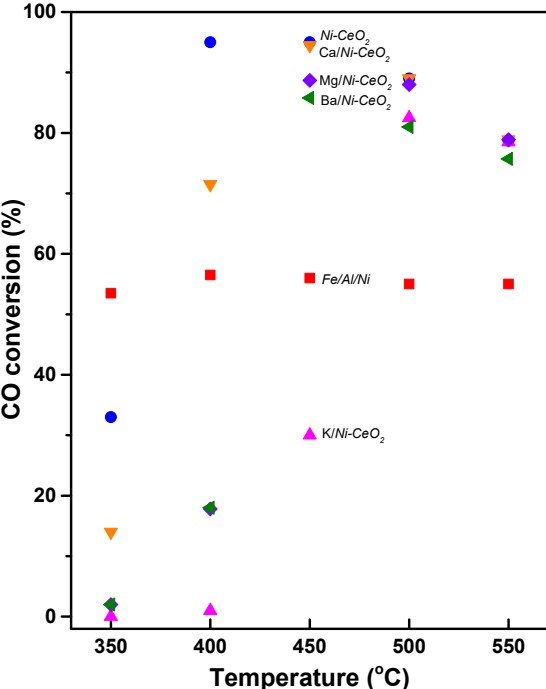

**Figure 7.** CO conversion of Ni-based catalysts.

### 3.4. Co-Based Catalyst

As discussed in the previous section (Ni), Co is also one of the non-precious candidates for the WGS reaction. A number of active sites and oxygen storage capacity are the key properties of the Co/CeO$_2$ catalyst. Research has tried to improve the characteristics of Co/CeO$_2$ by adding the promoter, optimizing the composition, and using various synthesis methods. In this section, four papers related to Co-based catalysts are summarized, and

descriptions are distinguished by naming [BCC], [CZC], [CC-M], [C-NC], and [CF] in front of descriptions for each characteristic analysis.

   **[BCC]** A catalyst study in which Ba was added to $Co/CeO_2$ was conducted. Catalysts were produced using CP and IWI (incipient wetness impregnation) method. The loading amount of Ba promoter in the 15 wt% $Co/CeO_2$ catalyst system was varied from 0, 1, 2, and 3 wt%, which were designated as $Co/CeO_2$ (BCC-0), 1% $BaCo/CeO_2$ (BCC-1), 2% $BaCo/CeO_2$ (BCC-2), and 3% $BaCo/CeO_2$ (BCC-3), respectively. Table 8 summarizes the physical properties of Co-based catalysts. The $Co/CeO_2$ catalyst with 1% Ba had a high BET surface area of 60 $m^2/g$ and a high surface $Co^0$ dispersion value [1]. Catalyst optimization studies using the $Co-CeO_2$ production technique revealed varied BET surface areas. When the BET surface area of the catalyst before and after the reaction was evaluated, it revealed a declining trend [6]. It was proven that there was no correlation between the BET surface area and the catalytic activity of the $Co-CeO_2$ catalyst related to the production process [6]. In $NbCo-CeO_2$ studies, where niobium oxide was added to $Co-CeO_2$, BET analysis was performed. As a result, $Co-CeO_2$ catalysts including Nb had equal BET surface areas and lower specific surface areas than catalysts containing no Nb [53]. **[CZC]** A catalyst study in the $Co-CeO_2$ catalyst promoted with $ZrO_2$. The $Co-CeO_2$ (C-C), $Co-ZrO_2$ (C-Z), and $Co-Zr_{(1-x)}Ce_{(x)}O_2$ catalysts were prepared using a previously reported co-precipitation method [8]. The Co content of all the catalysts was fixed at 15 wt%. The $CeO_2:ZrO_2$ molar ratio of the $Co-Zr_{(1-x)}Ce_{(x)}O_2$ catalysts was varied from $ZrO_2:CeO_2$ = 2:8, 4:6, 6:4, and 8:2, which were designated as $Co-Zr_{0.2}Ce_{0.8}O_2$ (CZ2C8), $Co-Zr_{0.4}Ce_{0.6}O_2$ (CZ4C6), $Co-Zr_{0.6}Ce_{0.4}O_2$ (CZ6C4), and $Co-Zr_{0.8}Ce_{0.2}O_2$ (CZ8C2), respectively. In a catalyst study in which $ZrO_2$ was introduced to $Co-CeO_2$, BET analysis revealed that as the Zr concentration rose, the BET surface area increased, but the catalytic activity remained unchanged [8]. **[CC-M]** $Co-CeO_2$ catalysts were prepared by various manufacturing methods such as sol-gel, incipient wetness impregnation, co-precipitation, and hydrothermal, which were designated as $Co-CeO_2$-SG (CC-SG), $Co-CeO_2$-IWI (CC-IWI), $Co-CeO_2$-CP (CC-CP), and $Co-CeO_2$-HT (CC-HT), respectively. Catalyst structure and oxygen vacancy concentration were probed by Raman spectroscopy. In the case of the CC-SG catalyst, this peak was clearly shifted to lower wave numbers, which was indicative of $CeO_2$ structural distortion that generated lattice strain and defects in the $CeO_2$ lattice, and thus promoted the formation of oxygen vacancies [6]. This indicates that the largest amount of oxygen vacancy was formed in the CC-SG catalyst. **[CF]** $CoFe_2O_4$ catalysts were prepared by various manufacturing methods such as electrospinning, sol-gel, hydrothermal, and co-precipitation, which were designated as $CoFe_2O_4$-ES, $CoFe_2O_4$-SG, $CoFe_2O_4$-HT, and $CoFe_2O_4$-CP, respectively. The $CoFe_2O_4$ (CP) catalyst had the largest BET surface area in a study on the production technique of $CoFe_2O_4$, and the BET surface area declined in the order of $CoFe_2O_4$ (CP) > $CoFe_2O_4$ (HT) > $CoFe_2O_4$ (ES) > $CoFe_2O_4$ (SG) [88]. In the catalyst study using the $CoFe_2O_4$ production process, the XRD analysis revealed the same $CoFe_2O_4$ peak [88]. **[C-NC]** Nb-doped $Co-CeO_2$ catalysts with 15 wt%. Co were prepared using a co-precipitation method. The Nb-doped $Co-CeO_2$ catalysts were denoted xNbCo, with "x" representing the weight percentage of 0.5 wt% Nb, 1.5 wt% Nb, and 2.5 wt% Nb. In addition, 0.5 wt% Nb-doped $Co-CeO_2$ catalysts were prepared by various manufacturing methods such as co-precipitation, incipient wetness impregnation, sol-gel, and hydrothermal, which were designated as Co-NC-CP (CNC-CP), Co-NC-IWI (CNC-IWI), Co-NC-SG (CNC-SG), and Co-NC-HT (CNC-HT), respectively. The BET analysis revealed surface area values in the order of Co-NC > Cu-NC > Fe-NC > Zn-NC in the metal material optimization study utilizing $Nb-CeO_2$ as a support. The Co-NC catalyst had the largest BET surface area value, and its increased surface area enhanced the mass transfer coefficient of reactants and products. Consequently, Co-NC and Cu-NC catalysts were expected to perform better than Fe-NC and Zn-NC catalysts [89]. The Co-NC-CP catalyst had the largest BET surface area in the optimization study of the Co-NC manufacturing process, and the value decreased in the order of Co-NC-CP > Co-NC-HT > Co-NC-IM > Co-NC-SG. CO chemisorption analysis revealed that the Co dispersion of the Co-NC catalyst decreased in the following order:

Co-NC-HT > Co-NC-CP > Co-NC-SG > Co-NC-IM. Co dispersion impacts catalytic activity because it is proportional to the number of active species in contact with the reactants [89]. There was a difference in lattice parameter values between $CeO_2$ and the manufactured catalyst in a metal material optimization study utilizing $Nb-CeO_2$ as a support owing to the difference in ionic radii between the active material and Ce ions [89]. Following the metal material optimization study using $Nb-CeO_2$ as a support, the production technique of the Co-NC catalyst with high activity was optimized. All catalysts had a $CeO_2$ peak, but only catalysts produced with SG and IM had a broad and small Co peak at 44.5 °C [89]. Table 9 shows the TPR findings of the Co-based catalyst. **[BCC]** All $Co/CeO_2$ catalysts with 1 wt% Ba added decreased at a lower temperature than other catalysts in a study in which Ba was added to $Co/CeO_2$ catalysts. The reducibility of cobalt oxide improved as the amount of Ba rose, as did the reduction temperature. This is due to the fact that the interaction between $CeO_2$ and Co was diminished with the addition of Ba [1]. The TPR analysis performed in the catalyst optimization study using the $Co-CeO_2$ production process revealed two primary temperatures, as shown in Table 9. At the largest temperature, the $Co-CeO_2$ catalyst produced with SG is reduced from CoO to $Co^0$. Consequently, CoO, an active species, was generated at a higher temperature in the HTS reaction than other catalysts, and it was hypothesized that the interaction between $Co^0$ and $CeO_2$ may be enhanced [6]. The $Co-CeO_2$ catalyst with Nb moved to a lower temperature than the $Co-CeO_2$ catalyst in the $H_2$-TPR data from the $NbCo-CeO_2$ study, in which niobium oxide was added to $Co-CeO_2$. Because of the addition of Nb, the concentration of electron mediators rose as $Ce^{4+}$ was replaced by $Nb^{5+}$, causing structural deformation of $CeO_2$ and increased oxygen defects. Because high OSC demonstrates high activity in the WGS process, the catalyst with Nb added was predicted to display better catalytic activity than the $Co-CeO_2$ catalyst. However, the reduction temperature of $Nb_2O_5$ was not detected, which seems to be due to the requirement for a high temperature of 900 °C or more [53]. **[CZC]** A study of catalysts containing $ZrO_2$ and $Co-CeO_2$ revealed four reduction temperatures. The first reduction temperature was created by $Co_3O_4$ reduction with large crystals, while the second reduction temperature was caused by $Co_3O_4$ reduction with small crystals to CoO. The third reduction temperature was the result of CoO to $Co^0$ reduction, while the fourth reduction temperature was thought to be a broad reduction temperature range created by bulk $CeO_2$ reduction [8]. **[CC-M]** The reduction at 211–327 °C was attributed to the reduction of $Co_3O_4$ to CoO, while the second reduction at 262–414 °C represented the reduction of CoO to $Co^0$. The temperature of the $Co_3O_4$ to CoO increased in the order of CC-HT < CC-SG < CC-CP < CC-IWI, while that of the CoO to $Co^0$ increased in the order of CC-HT < CC-CP < CC-IWI < CC-SG. The reduction of CoO to the $Co^0$ in CC-SG catalyst occurred at the highest temperature. This means that the active species in HTS, $Co^0$ (metallic cobalt), is formed at a higher temperature than other catalysts in the case of the CC-SG catalyst. **[C-NC]** $H_2$-TPR analysis of each catalyst was performed in the metal optimization study utilizing $Nb-CeO_2$ as a support. The first temperature was the reduction temperature of $Co^{3+}$ to $Co^{2+}$ in the $H_2$-TPR analysis of the Co-NC catalyst, the second temperature was the reduction temperature of $Co^{2+}$ to metallic Co, and the third temperature was the reduction temperature of bulk $CeO_2$ on the catalyst reduction temperatures, a key feature in the oxidation–reduction process in the WGS surface. The first temperature in the Cu-NC catalyst's $H_2$-TPR analysis was the reduction temperature from $Cu^{2+}$ to $Cu^+$, while the second temperature was the reduction temperature from $Cu^+$ to metallic Cu. Lower reaction and the reduction temperature increased in the order Cu-NC < Co-NC < Fe-NC < Zn-NC [89]. The reduction temperatures of the manufactured catalysts displayed distinct patterns as a consequence of $H_2$-TPR in the Co-NC manufacturing technique optimization study. This suggests that the preparation process influences the reducibility of the Co-NC catalyst. The reduction temperature increased in the order of Co-NC-SG < Co-NC-IM < Co-NC-CP < Co-NC-HT [89]. **[CF]** Table 10 shows the TPR findings of the $CoFe_2O_4$ catalyst. Only in the case of the catalyst manufactured by electrospinning did a distinct reduction peak occur at a temperature of around 300 °C as a result of $H_2$-

TPR analysis in the catalyst study according to the production technique of $CoFe_2O_4$. Similar findings have been reported for nanowire-structured $CoFe_2O_4$ spinel catalysts. Complex overlapping peaks emerged owing to iron reduction ($Fe_3O_4$ to FeO; FeO to $Fe^0$), and Co appeared at temperatures over 400 °C. Catalysts generated by means other than electrospinning exhibited overlapping peaks, including conversion of $Fe_2O_3$ to $Fe_3O_4$ [88]. **[BCC]** As a consequence of the catalytic reaction, the $Co/CeO_2$ catalyst with 1 wt% Ba showed the largest CO conversion rate in the catalyst study in which Ba was added to the $Co/CeO_2$ catalyst. It had the maximum stability and catalytic activity in the stability test [1]. Figure 8 shows CO conversion results. The reaction of the catalyst synthesized with SG demonstrated the largest CO conversion rate (90%) in the catalyst optimization study according to the production technique of the $CeO_2$ catalyst. Furthermore, all other catalysts, with the exception of the hydrothermal synthesis catalyst, did not create side reactions [6]. The catalytic reaction of $NbCo-CeO_2$ with niobium oxide added to $Co-CeO_2$ exhibited high CO conversion in the descending order of 1.5 wt% $NbCo-CeO_2$ > 0.5 wt% $NbCo-CeO_2$ > 2.5 wt% $NbCo-CeO_2$ > $Co-CeO_2$. Even at high space velocity conditions, the 1.5 wt% $NbCo-CeO_2$ catalyst demonstrated good activity and stability with no side reactions [53]. **[CC-Z]** At 450 °C, the catalysts in which $ZrO_2$ was added to $Co-CeO_2$ decreased in the order of $Zr_{0.6}Ce_{0.4}O_2$ > $Zr_{0.4}Ce_{0.6}O_2$ > $Zr_{0.2}Ce_{0.8}O_2$ > $Zr_{0.8}Ce_{0.2}O_2$ > $CeO_2$ > $ZrO_2$. All catalysts demonstrated comparable CO conversion rates at reaction temperatures of 500 °C and 550 °C. High CO conversion rates were observed for $Co-Zr_{0.6}Ce_{0.4}O_2$, $Co-Zr_{0.4}Ce_{0.6}O_2$, and $Co-Zr_{0.2}Ce_{0.8}O_2$ catalysts with high OSC values. However, the $Co-Zr_{0.4}Ce_{0.6}O_2$ catalyst with the highest OSC showed a lower CO conversion rate than the $Co-Zr_{0.6}Ce_{0.4}O_2$ catalyst. This result may be attributable to the $Co-Zr_{0.6}Ce_{0.4}O_2$ catalyst's easier reducibility [8]. **[CF]** Only the catalyst generated by electrospinning had a high CO conversion rate of more than 80% during the HTS process; whereas, other catalysts showed low CO conversion rates. Furthermore, only the catalyst produced by electrospinning did not exhibit side reactions in all ranges. Catalysts other than those generated by electrospinning seem to have low oxygen vacancy, making formation of an active phase problematic [88]. **[C-NC]** The Co-NC catalyst demonstrated high CO conversion after 450 °C in a metal optimization study utilizing $Nb-CeO_2$ as a support. It had a 100% $CO_2$ selectivity, and no side reactions occurred. The Co-NC catalyst demonstrated a consistent conversion rate after 60 h of reaction with the Cu-NC catalyst. Among the prepared catalysts, the Co-NC catalyst was the most active [89]. The catalyst synthesized with CP had the largest oxygen vacancy concentration in the study of optimizing the production technique of Co-NC. Because of the high OSC and Co dispersion, the maximum CO conversion rate was higher than 500 °C. Furthermore, the high OSC and robust contact between the Co and NC supports resulted in consistent Co-NC-CP catalytic activity [89]. As a reaction result of the Co-based catalyst, catalyst activity was measured in a various range of GHSV. The $CoFe_2O_4$-ES catalyst showed relatively high catalytic activity at the GHSV of 44,500 $h^{-1}$. The high catalytic activity of the $CoFe_2O_4$-ES catalyst is due to its superior redox property. This superior redox property may easily induce the formation of an active phase ($Co^0$ and $Fe_2O_3$) in the $CoFe_2O_4$.

**Table 8.** Characteristics of Co-based catalysts.

| Catalysts | BET Surface Area ($m^2/g$) [a] | $Co^0$ Crystallite Size (nm) [b] | Lattice Parameter (A) [b] | Co Dispersion (%) [c] | $Co^0/$ ($Co^0 + Co^{2+} + Co^{3+}$) (%) [d] | Ref. |
|---|---|---|---|---|---|---|
| BCC-1 | 60 | N.A. [a] | - | 0.63 | 49.8 | [1] |
| CC-SG | 30 | - | - | 1.61 | - | [6] |
| 1.5NbCo | 114 | 4.7 | 5.430 | 3.47 | 26 | [53] |
| CZ6C4 | 186.8 | 8.7 | 5.380 | 1.96 | 42.3 | [8] |
| $CoFe_2O_4$-ES | 5.8 | 35.3 | - | - | 34.4 | [88] |
| CNC-CP | 115.39 | - | 5.428 | 3.41 | 50.80 | [89] |

[a] Estimated from $N_2$ adsorption isotherm at −196 °C. [b] Calculated from XRD. [c] Estimated from the CO chemisorption results. [d] Estimated from the CO 2p XPS profiles.

**Table 9.** Reduction characteristics of Co-based catalysts.

| Catalyst | $Co_3O_4$ to CoO (°C) | CoO to $Co^0$ (°C) | Surface Oxygen Species of $CeO^2$ (°C) | Ref. |
|---|---|---|---|---|
| BCC-0 | 327 | 376 | - | |
| BCC-1 | 264 | 333 | - | |
| BCC-2 | 285 | 354 | - | [1] |
| BCC-3 | 299 | 390 | - | |
| CC-SG | 224 | 414 | - | |
| CC-IWI | 327 | 377 | - | [6] |
| CC-CP | 295 | 367 | - | |
| CC-HT | 211 | 262 | - | |
| CZ2C8 | 335 | 430 | - | |
| CZ4C6 | 326 | 411 | - | |
| CZ6C4 | 335 | 420 | - | [8] |
| CZ8C2 | 326 | 411 | - | |
| C-Z | 308 | 353 | - | |
| C-C | 320 | 414 | 573 | |
| 0.5NbCo | 304 | 414 | 532 | |
| 1.5NbCo | 304 | 414 | 532 | [53] |
| 2.5NbCo | 304 | 414 | 532 | |
| CNC-CP | 304 | 414 | 532 | |
| CNC-IWI | 333 | 385 | 424 | |
| CNC-SG | 302 | 354 | 491 | [89] |
| CNC-HT | 322 | 436 | 768 | |

**Table 10.** Reduction characteristics of $CoFe_2O_4$ catalysts.

| Catalyst | $Fe_2O_3$ to $Fe_3O_4$ (°C) | Ref. |
|---|---|---|
| $CoFe_2O_4$-ES | 351 | |
| $CoFe_2O_4$-SG | 558 | [88] |
| $CoFe_2O_4$-HT | 598 | |
| $CoFe_2O_4$-CP | 544 | |

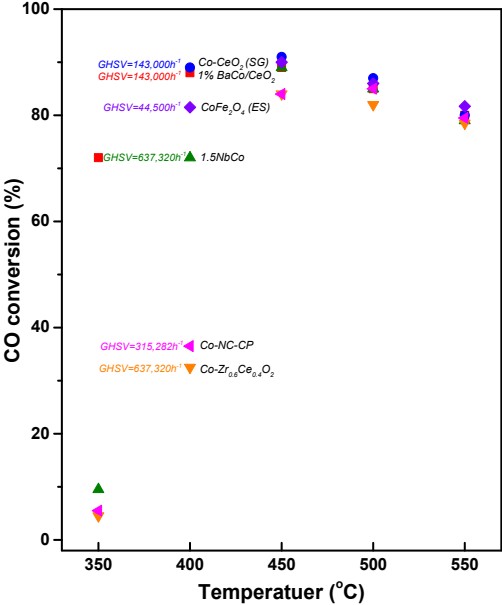

**Figure 8.** CO conversion of Co-based catalysts.

### 3.5. Pt-Based Catalyst

Various precious metals were applied for the WGS reaction, but Pt-based catalysts were known to be useful for the WGS reaction of waste synthesis gas because of their

excellent activity and operation in a wide range of temperatures. In addition, Pt-based catalysts have been reported to have a high activity even in the presence of impurity ($H_2S$), which typically exists in the renewable resources. In this section, two papers related to Pt-based catalysts are summarized, and descriptions are distinguished by naming [Pt-S] and [Pt-L] in front of descriptions for each characteristic analysis.

**[Pt-S]** Various supports were prepared by precipitation, and $Pt/CeO_2$, $Pt/MgO$, $Pt/Al_2O_3$, and $Pt/ZrO_2$ catalysts were produced using the IWI method. The loading amount of Pt on various supports was fixed at 2 wt% [59]. **[Pt-L]** A catalyst study in which Pt was added to $CeO_2$ was conducted. Catalysts were produced using precipitation and the IWI method.

The loading amount of Pt in the $CeO_2$ supports varied at 0.1 wt%, 0.5 wt%, 2.0 wt%, 5.0 wt%, and 10.0 wt%. Table 11 summarizes the physical properties of Pt-based catalysts [57]. **[Pt-S]** The $Pt/CeO_2$ catalyst had the lowest surface area value of 77 $m^2/g$ as a result of BET analysis in the study of Pt catalysts utilizing different supports. Despite having the smallest surface area, the degree of $Pt^0$ dispersion was the most significant. This seems to be related to the high amount of deficient oxygen in $CeO_2$ [59]. **[Pt-L]** According to BET analysis in a study where $Pt/CeO_2$ catalysts were made by changing the amount of Pt, the surface area when Pt was added decreased compared to $CeO_2$. Among the $Pt/CeO_2$ catalysts, the catalyst with 2 wt% Pt added had the largest surface area [57]. Table 12 shows the TPR results of the Pt-based catalyst. **[Pt-S]** Only the $Pt/CeO_2$ catalyst consumed a substantial amount of hydrogen at less than 100 °C as a result of $H_2$-TPR analysis in the study of Pt catalysts employing different supports. $PtO_x$ species with weak interactions with the support seem to be diminished even at room temperature [59]. **[Pt-L]** Two reduction temperatures were found in the $H_2$-TPR analysis results for all catalysts in the study of synthesizing $Pt/CeO_2$ catalysts by altering the amount of Pt. The first temperature was the reduction temperature of $PtO_x$ to metal $Pt^0$, while the second temperature was the bulk $CeO_2$ reduction temperature. Among the produced catalysts, the 2% $Pt/CeO_2$ catalyst had the lowest reduction temperature and the highest hydrogen consumption at around 75 °C. As reverse spillover of $CeO_2$ lattice oxygen developed at a low temperature, the second reduction temperature of the 2% $Pt/CeO_2$ catalyst diminished. When $CeO_2$ and Pt-added $CeO_2$ are compared, it can be concluded that adding Pt causes oxygen reverse spillover [57]. Figure 9 shows CO conversion results. As a result of a Pt-based catalyst reaction, Pt catalysts were tested in reaction using various supports. **[Pt-S]** Because the concentration of sulfur in waste-derived syngas varies greatly, the stability test was carried out while varying the quantity of $H_2S$ from 0 to 1000 ppm. When $H_2S$ was supplied at less than 100 ppm, thermodynamic equilibrium was virtually attained without deactivation for over 100 h. The catalytic activity decreased when $H_2S$ was introduced at 500 and 1000 ppm, although it remained above 60% for 100 h. When the $H_2S$ infusion was halted, the catalytic activity was totally recovered. Catalysts supported by $CeO_2$ showed good oxidation–reduction characteristics. The $Pt/CeO_2$ catalyst performed best in terms of activity and sulfur tolerance. Furthermore, it demonstrated a remarkable regeneration rate owing to high OSC and persistent catalytic activity even when $H_2S$ concentrations were raised to 1000 ppm [59]. **[Pt-L]** Reactions were performed to demonstrate sulfur resistance and catalytic activity in a study in which $Pt/CeO_2$ catalysts were produced by changing the amount of Pt. As a consequence, the catalyst with a $Pt/CeO_2$ content of 2% demonstrated good initial CO conversion, sulfur resistance, and regeneration. When exposed to 500 ppm $H_2S$, catalysts with 0.1 and 0.5 wt% Pt demonstrated low sulfur tolerance. Catalysts with 5 and 10% wt% Pt included a significant amount of scattered $Pt^0$. However, since the amount of Pt is substantial, the particle size may be rather large, and this has the disadvantage that it is easy to sinter during high-temperature reactions. In conclusion, the 2 wt% $Pt/CeO_2$ catalyst is best suited as a sulfur-resistant catalyst for the practical use of waste-derived syngas [57]. As a result of the reaction of the Pt-L catalysts, the 2 wt% $Pt/CeO_2$ catalyst showed the highest catalytic activity. Among prepared Pt-L catalysts, the 2 wt% $Pt/CeO_2$ catalyst demonstrated the best redox characteristics, with the highest number of $O_v$ and

$Ce^{3+}$ species related to oxygen vacancy, the highest OSC, and the most dispersed $Pt^0$ species. In the Pt-S catalyst, the reaction between sulfur adsorbed to Pt by the $Pt/CeO_2$ catalyst and the mobile oxygen generated by the $CeO_2$ support influenced the regeneration mechanism and had high sulfur tolerance.

**Table 11.** Characteristics of Pt-based Catalysts.

| Catalysts | BET Surface Area ($m^2$/g) | $Pt^0$ Dispersion (%) | OSC ($10^{-4}$ $gmol/g_{cat}$) | Ref. |
|---|---|---|---|---|
| $Pt/CeO_2$ | 77 | 76.29 | 6.66 | [59] |
| $Pt/ZrO_2$ | 284 | 59.14 | 2.04 | [59] |
| $Pt/MgO$ | 167 | 76.18 | 0.86 | [59] |
| $Pt/Al_2O_3$ | 202 | 61.10 | 1.87 | [59] |
| $CeO_2$ | 105 | - | 3.93 | [57] |
| 0.1 wt% $Pt/CeO_2$ | 62 | 94.1 | 6.30 | [57] |
| 0.5 wt% $Pt/CeO_2$ | 66 | 81.3 | 6.46 | [57] |
| 2.0 wt% $Pt/CeO_2$ | 77 | 76.3 | 6.66 | [57] |
| 5.0 wt% $Pt/CeO_2$ | 73 | 38.1 | 6.34 | [57] |
| 10.0 wt% $Pt/CeO_2$ | 56 | 10.5 | 5.27 | [57] |

**Table 12.** Reduction characteristics of Pt-based Catalysts.

| Catalyst | $PtO_x$ to $Pt^0$ (°C) | Bulk $CeO_2$ (°C) | Ref. |
|---|---|---|---|
| $Pt/CeO_2$ | 75 | - | |
| $Pt/ZrO_2$ | 426 | - | [60] |
| $Pt/MgO$ | 261 | - | |
| $Pt/Al_2O_3$ | 80 | - | |
| $CeO_2$ | - | 774 | |
| 0.1% $Pt/CeO_2$ | 238 | 758 | |
| 0.5% $Pt/CeO_2$ | 129 | 758 | [57] |
| 2.0% $Pt/CeO_2$ | 75 | 532 | |
| 5.0% $Pt/CeO_2$ | 120 | 752 | |
| 10.0% $Pt/CeO_2$ | 155 | 752 | |

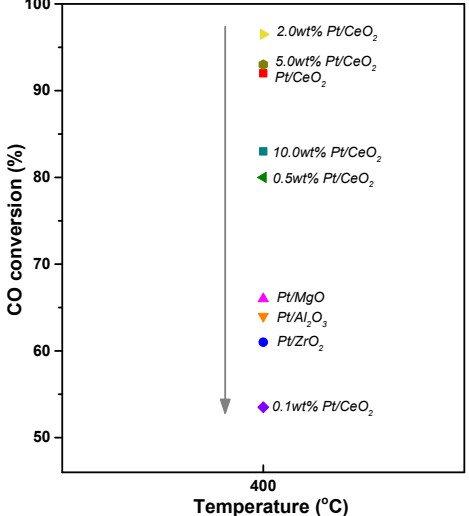

**Figure 9.** CO conversion of Pt-based Catalysts.

### 4. Conclusions and Perspectives

The waste policies of many countries have focused on energy recovery in line with the so-called "waste-to-energy" (WtE) strategy. In particular, it is very useful to produce hydrogen from a product that is obtained from a thermal process of waste such as pyrolysis and gasification (excluding incineration). However, the primary issues are a wide range of product composition and high concentrations of CO, which in turn lead to the restriction of their upgrade to hydrogen via the WGS reaction. Therefore, research has continuously studied the improvement of catalytic performance for the WGS reaction. In this review, we discussed the characterization and catalytic performance of Fe-, Cu-, Ni-, Co-, and Pt-based catalysts for the WGS reaction in order to provide a guideline for designing an appropriate catalyst. It was necessary to obtain the catalysts having the largest number of active sites in common. In addition, the redox ability was the one of the core characteristics, considering the redox mechanism of the WGS reaction. The detailed results depending on the composition of catalysts are as follows: First, the easier reducibility was one of the most significant properties for Fe-based catalysts. The rapid redox cycle between $Fe^{2+}$ and $Fe^{3+}$ mainly determined the catalytic activity for the WGS reaction. Cu played a key role in enhancing the reducibility of Fe and provided additional active sites. In addition, a harmless textural promoter, such as Al, was necessary for Fe-based catalysts which were prone to be a sintering. Second, Cu dispersion and oxygen storage capacity importantly affected the catalytic activity of $Cu/CeO_2$ and promoted catalysts. The strong interaction between CuO and $CeO_2$ was related to the redox cycle of $Cu^{2+} \leftrightarrow Cu^{1+}$ and $Ce^{4+} \leftrightarrow Ce^{3+}$. Third, it was of paramount importance to prevent the occurrence of methanation reaction over Ni- and $Ni/CeO_2$-based catalysts, apart from enhancing the dispersion and oxygen storage capacity. Alkali or alkali-earth metals were effective for the suppression of methanation reactions. Fourth, the amounts of Co active sites and oxygen storage capacity were significant for $Co/CeO_2$ and promoted catalysts, which was significant for the $Cu/CeO_2$ catalyst. Fifth, the resistance against impurity was one of the key parameters because of the importance and practicality of the renewable resources. The $Pt/CeO_2$ catalyst showed excellent activity with high resistance to $H_2S$ even at a 550 ppm of $H_2S$. A lot of breakthrough research in this area will provide appropriate directions for the improvement of FWGS catalysts in order to establish the industrialization of hydrogen production processes from waste. Fe-Al-Cu catalysts with improved reducibility and the addition of new active sites are promising to replace a commercial Fe-Cr catalyst as the WGS catalyst for hydrogen production from waste. In addition, it is reasonable to combine the Ni and Co active metals with a $CeO_2$ support which has a high OSC. These non-noble metals-based catalysts are useful and economically feasible, but they are active under the conditions where impurities are eliminated. $Pt/CeO_2$ only exhibits excellent catalytic performance even under conditions containing impurities, thus it can be useful for a small-scale facility which restricts the installation of processes to eliminate impurities.

**Author Contributions:** Writing—original draft, R.-R.L. and I.-J.J.; conceptualization, J.-O.S., W.-J.J. and H.-S.R.; supervision, J.-O.S., W.-J.J. and H.-S.R. All authors have read and agreed to the published version of the manuscript.

**Funding:** This research received no external funding.

**Institutional Review Board Statement:** Not applicable.

**Informed Consent Statement:** Not applicable.

**Acknowledgments:** This paper was supported by Wonkwang University in 2021.

**Conflicts of Interest:** The authors declare no conflict of interest.

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
