# Peer review of "Advances in Catalysts for Water–Gas Shift Reaction Using Waste-Derived Synthesis Gas"

_catalysts, doi:10.3390/catal13040710_

Round 1
Reviewer 1 Report
In recent years, with the growth of population and the expansion of urbanization and industrialization, the amount of waste discharged each year has increased year by year. Therefore, the production of clean energy H2 by water gas conversion reaction (WGS) of waste is a potential waste utilization method, which is conducive to environmental protection and sustainable development. The author introduces the water-gas shift reaction in detail, and systematically summarized the core characteristics of Fe, Cu, Ni, Co and Pt in WGS reaction that are favorable for the reaction. In addition, the author also introduced the support of Al2O3 and CeO2 for WGS catalyst. The work deserves to be published, but I have a few small questions and would appreciate a response. Below is the summary of my major concerns.
1. In this paper, the progress of Fe-based catalysts, Cu-based catalysts, Ni-based catalysts and Pt-based catalysts in WGS reaction reported in the literature is summarized in detail. However, the author should also summarize the decisive role of the added additives or metal types for each catalyst, which will be more conducive to guiding the synthesis of high-performance WGS catalysts.
2. Ni-based catalysts are suitable for HT-WGS reaction, but CO2 and H2 in the product will also trigger RWGS reaction, how to avoid it?
3. Some legends in the text are not very clear. Please provide high-definition pictures, such as Figure 2 and Figure 3, and please confirm the format of Figure 2.
Reviewer 2 Report
The paper written by Ru-Ri Lee and co-workers entitled ,,Advances in Catalysts for Water-Gas Shift Reaction Using Waste-derived Synthesis Gas is interesting and may expand the existing knowledge concerning the water gas shift reaction, but at this moment this work cannot be published in a current form. Below I am presenting selected comments that come to my mind after reading this article.
In its current form, the work is difficult for the reader to understand, some of the comments and questions that have been raised over after reading the work:
- Figures 3 and 4 need to be corrected, they are currently illegible. All data presented in the figures should be clearly described.
- Descriptions containing the names [NCC], [CCA], [CBCA] [NCC], [CCA], and [CBCA]etc. should be clearly explained in the text of the manuscript in order to better understand the text of the manuscript.
- When compared to other catalysts, the Fe/Al/Cu catalyst achieved the maximum CO conversion and 100% CO2 selectivity. – What does it mean ? compared to what catalysts? And at low or high temperature?
- The easy reducibility of Fe2O3, the synergistic effect of Cu and Al, and the high stability of the Fe/Al/Cu catalyst all contribute to the high activity of catalyst [12].- Could the authors explain this statement in detail?
- Finally, a FAC-PC-3-240 catalyst was developed with a 40-fold increase in production of the FAC-PC-3 catalyst ramped up three times.- Could the authors explain this statement in detail?
- The authors describe in the text ,,Catalyst characterization revealed the high activity of the Fe/Al/Cu catalyst, which is presented in Table 2. According to BET analysis result, the greater the surface area and the smaller the size of the Fe3O4 crystallites, the greater the activity of the catalyst’’ – but Table 2 does not contain any activity results.
- Furthermore, when FAC catalysts were made, the FAC-CP catalysts pro-duced with CP had the maximum surface area and the lowest Fe3O4 crystallite size- I suggest correcting the sentence.
- Abbreviation a is related to the table 2 concern the BET values not the XRD measurements??
- When the Fe/Al/Cu catalyst was grown up three times, the FAC-PC-3 catalyst had a surface area and Fe3O4 crystallite size equal to the current FAC-PC-1 catalyst – I do not undesrsood this sentence.
- Table 2 does not contain any results describing Cu dispersion.
- FAC-PC-3 – a detailed description of abbreviations must be introduced in the work to facilitate understanding of the work.
- In the WGS reaction, the production of γ-Fe2O3 is critical- what does mean this statement?
- The reduction mechanism of the e catalysts should be analyzed again. FeO can not be the final product of the reduction process of the Fe based system.
- The first reduction temperature is caused by a decrease of CuO species- what does it mean?
- (Fe8/3+) ??
- It was confirmed as a result of this that reduction happens more readily owing to the synergistic effect of Fe/Al and Cu, a metal oxide [12] – I do not understand this statement.
- Similar to the aforementioned findings, the FAC-CP catalyst made from CP – I think this system is prepared using CP method.
- Among several production methods, the FAC-CP catalyst has the lowest reduction temperature and is easily reducible - I do not understand this statement.
- Pure Fe has three reduction temperatures of 405, 700, and 950 °C, where α-Fe2O3 is reduced to Fe3O4, Fe3O4 to FeO, and FeO to Fe0, respectively - I do not understand.
- FAC catalyst was likewise shifted to a lower temperature than the pure Fe catalyst – I do not understand, maybe the reduction profiles ?
- This suggests that the presence of Al and Cu facilitates the formation of the Fe3O4 active phase – but how???
- The description of the XPS results presented in Figure 2 is hard to understand because the authors did not provide the catalysts treatment for the presented spectra. Have the catalysts been calcined or reduced?
- When the Fe/Al and metal oxide doped catalysts were reacted at a determined by calculating the area of the peak, and 47% of reduced Cu species were found on the surface of the FAC-CP catalyst.- sentence should be rewritten
- ,,Cu metal provides active oxygen species to oxidize’’ - but how?
- Furthermore, the Fe/Al/Cu catalyst demonstrated 100% CO2 selectivity and 0% CH4 selectivity for CO2 and CH4 selectivity, as well as very stable CO conversion after a 100-hour stability test at 400 °C - sentence should be rewritten
- All FAC catalysts produced using different synthesis methods showed 100% CO2 selectivity and 0% CH4 selectivity, which was consistent with prior study findings. - sentence should be rewritten
- The CoFe2O4-MA catalyst has a Co3O4 spinel structure with the lowest crystallite size and a low intensity peak (3.7 nm) – but in table is presented the result of the crystallite size of spinel ferrite
- A great deal of research is underway to introduce various supports into Cu - sentence should be rewritten
- Catalysts were created in a variety of methods in order to determine the best approach for creating the Ni-Cu-CeO2 catalyst - - sentence should be rewritten
- The description of the results presented in table 2 should be corrected.
- How was the size of the CeO2 crystallites determined in Table 5?
- The description of the reduction stages observed for the copper catalysts presented in Table 6 should be rewritten because at this moment it is hard to understand the reduction stages e.g. the first effect is described following: CuO ↔ CeO2 (oC). What does mean? This scheme present the first reduction stage??
- It is hard to understand the XPS results without the presented data.
- The paragraph concerning Ce/Cu/γ-Al2O3 catalysts doped by Ba, Zr, and Nd should be rewritten to better understand the presented data.
- Figure 6 should be divided into two figures in order to better visualization the presented data.
- What does it mean the words Fresh Fe2O3 and Used Fe3O4 in Table 7?
- Table 8 should be corrected
- The description of the results presented in Figure 8 should be corrected for better visualization of the results.
- The description of the results presented in Figure 9 should be corrected for better visualization of the results.
- Catalyst study in which Pt was added to CeO2, MgO, Al2O3, and ZrO2 was conducted- I do not understand.
- What does it mean IWI method?
Due to all these problems, I would recommend the article for publication after major revision.
Reviewer 3 Report
In this review, different types of the catalysts developed in the past decade based on Fe, Cu, Ni, Co and Pt were summarized, and discussed in detail for WGS reaction, their characteristics were compared, and their limitations and future perspectives were also discussed. The literatures were up to date, and covered a broad scope of the recent research focusing on the catalyst design and optimization, and closely related to the central topic of the review. I recommend publication in the current format.
Round 2
Reviewer 2 Report
I think that the article can be accepted in its current form for publication in the Catalysts journal.